# Just-In-Time Reinforcement Learning: Continual Learning in LLM Agents Without Gradient Updates

Yibo Li [1]   Zijie Lin [1]   Ailin Deng [1]   Xuan Zhang [1]   Yufei He [1]   Shuo Ji [1]   Tri Cao [1]   Bryan Hooi [1]

## Abstract

While Large Language Model (LLM) agents excel at general tasks, they inherently struggle with continual adaptation due to the frozen weights after deployment. Conventional reinforcement learning (RL) offers a solution but incurs prohibitive computational costs and the risk of catastrophic forgetting. We introduce **Just-In-Time Reinforcement Learning (JitRL)**, a training-free framework that enables test-time policy optimization without any gradient updates. JitRL maintains a dynamic, non-parametric memory of experiences and retrieves relevant trajectories to estimate action advantages on-the-fly. These estimates are then used to directly modulate the LLM's output logits. We theoretically prove that this additive update rule is the exact closed-form solution to the KL-constrained policy optimization objective. Extensive experiments on WebArena and Jericho demonstrate that JitRL establishes a new state-of-the-art among training-free methods. Crucially, JitRL outperforms the performance of computationally expensive fine-tuning methods (e.g., WebRL) while reducing monetary costs by over $30\times$, offering a scalable path for continual learning agents. The code is available at https://github.com/liushiliushi/JitRL.

## 1. Introduction

While humans can learn "on the fly", current AI agents fundamentally lack this ability because their weights are frozen after training (Gao et al., 2025; Fang et al., 2025; Li et al., 2025b). This limitation severely restricts their practicality: when deployed in unfamiliar or dynamic settings, AI agents are analogous to newly hired employees without on-the-job training, which often repeatedly make the

[1]National University of Singapore, Singapore. Correspondence to: Shuo Ji <jishuo@u.nus.edu>.

*Proceedings of the 43rd International Conference on Machine Learning*, Seoul, South Korea. PMLR 306, 2026. Copyright 2026 by the author(s).

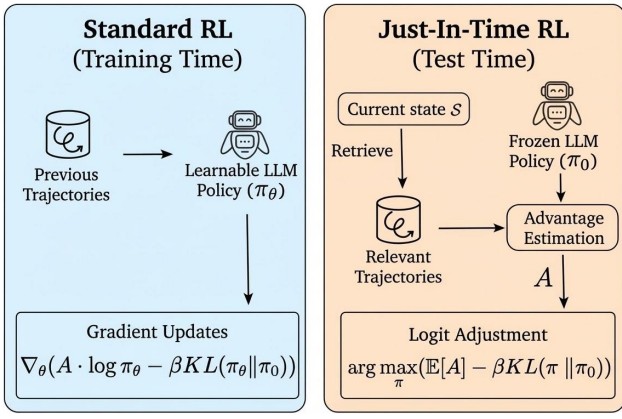

*Figure 1.* While standard RL performs policy gradient updates during training using previous trajectories, JitRL operates at test time. Specifically, it retrieves trajectories relevant to the current state to estimate advantages $A$, subsequently refining the output logits through a KL-regularized policy optimization objective.

same errors due to their failure to learn, whereas humans can adapt to their environment, including by learning from their mistakes. Highlighting the importance of this issue, Hendrycks et al. (2025) proposes an "AGI Score" assessing progress toward artificial general intelligence and finds that current AI systems are most deficient in their "capability to continually learn new information".

One potential solution is conventional reinforcement learning (RL) (Qi et al., 2025; Qian et al., 2025; Li et al., 2025a), performed in a continually policy gradient updating manner. However, this approach requires a large amount of training data to perform well. It is also computationally expensive, due to the high cost of RL and the need for frequent updates to keep the model up to date. In addition, it is prone to catastrophic forgetting, degrading the model's previously learned capabilities (Li et al., 2024). Prior work finds that conventional RL yields only limited improvements when evaluated in continual adaptation settings (He et al., 2025).

An alternative approach to this problem relies on in-context learning (ICL) (Dong et al., 2024) to enable the agent to learn at test time (Shinn et al., 2023; Wang et al., 2025). These methods rely on prompt or context engineering to incorporate information from past experiences. However,

such approaches tend to struggle as the required context length grows, especially since agentic tasks often involve long sequences of interactions (Liu et al., 2024). Crucially, ICL lacks the generality of RL. While prompts are confined to explicit textual descriptions, RL optimizes policies through rewards, enabling the mastery of complex skills that are difficult to articulate in text.

In light of these limitations, a fundamental question arises: *Can we enable agents to learn continually in a flexible way, without expensive parameter updates?* Our approach, Just-In-Time Reinforcement Learning (JitRL), enables general-purpose learning by adopting an RL formulation, while maintaining efficiency by avoiding gradient updates. Instead of gradient updates, JitRL maintains a dynamic memory bank that stores trajectories as `<state, action, reward>` triplets, as illustrated in Figure 1. Then, given the agent's current state, it retrieves relevant trajectories from the memory, and learns from them "just in time." To do so, it uses these trajectories to estimate the advantage of each action in the current state – i.e., how much better each action is relative to the average. These advantage estimates are then used to adjust the model's output logits. Crucially, we theoretically prove that our update rule is the exact closed-form solution to the policy optimization objective under a KL constraint. In this way, JitRL enables continual self-evolution without the prohibitive costs of gradient-based training.

We empirically validate JitRL on two benchmarks: WebArena (Zhou et al., 2024) for realistic web navigation and Jericho (Hausknecht et al., 2020) for long-horizon text-based games. Extensive experiments demonstrate that JitRL sets a new state-of-the-art, outperforming training-free baselines and weight-update methods, while reducing the monetary costs of conventional RL by over 30×. Furthermore, our results highlight JitRL's robustness, showing consistent gains across diverse LLM backbones and effective generalization to unseen tasks.

## 2. Related Work

**Reinforcement Learning.** Reinforcement learning (RL) algorithms like PPO (Schulman et al., 2017) and GRPO (Shao et al., 2024) have effectively aligned LLM agents for complex tasks. This paradigm spans diverse domains, ranging from optimizing search queries in information retrieval (Jin et al., 2025; Song et al., 2025) to refining tool execution (Li et al., 2025a; Qian et al., 2025). However, these gradient-based methods suffer from prohibitive computational costs and yield static models, fundamentally limiting their adaptability to distribution shifts.

**Training-Free Inference Enhancement.** Recent works have integrated external memory modules to enhance agents

directly at test time. Systems like MemGPT and Generative Agents (Packer et al., 2023; Park et al., 2023) utilize hierarchical memory to store historical interactions, while frameworks like Voyager and Reflexion (Wang et al., 2023; Shinn et al., 2023) retrieve textual descriptions of past skills or failures to improve future performance. Similarly, A-mem (Xu et al., 2025) constructs interconnected knowledge networks through dynamic indexing. Rather than merely retrieving text for in-context learning, our method treats memory as a non-parametric policy distribution. We perform soft updates directly on the LLM's logits, effectively achieving policy improvement without the overhead of parameter updates.

## 3. Preliminaries

Reinforcement Learning (RL) aims to optimize a policy $\pi_\theta$, which maps a given state to a probability distribution over actions, to maximize the expected cumulative reward. The objective function is defined as $J(\theta) = \mathbb{E}_{\tau \sim \pi_\theta}[R(\tau)]$, where $\tau = (s_0, a_0, r_0, s_1, a_1, r_1 \ldots, s_n, a_n, r_n)$ represents a trajectory of $n$ interactions, and $R(\tau)$ denotes the cumulative discounted return of that trajectory.

A fundamental approach to optimize this objective is the Policy Gradient method (Williams, 1992; Sutton et al., 1999). Intuitively, this method shifts the probability mass towards actions that yield higher returns by updating parameters in the direction of the gradient:

$$\nabla_\theta J(\theta) = \mathbb{E}_{s, a \sim \pi_\theta}[\nabla_\theta \log \pi_\theta(a|s) \cdot A(s, a)], \quad (1)$$

where $A(s, a)$ is the **advantage function**. The advantage function serves as a critic to evaluate the selected action relative to a baseline, formulated as:

$$A(s, a) = Q(s, a) - V(s). \quad (2)$$

Here, $Q(s, a)$ is the **action-value function**, representing the expected return of taking action $a$ in state $s$, while $V(s)$ is the **state-value function**, representing the average expected return of being in state $s$. Thus, $A(s, a)$ quantifies *how much better* a specific action is compared to the average performance of the policy in that state.

Reinforcement Learning typically requires the training of additional value networks to estimate the advantage. This process is computationally expensive and results in a static model that lacks the flexibility to adapt at test time.

## 4. Method

To address the limitations of gradient-based RL, we present **Just-In-Time Reinforcement Learning (JitRL)** framework. Instead of updating model parameters $\theta$, JitRL functions as a test-time policy optimization method that modulates a frozen prior $\pi_\theta$ towards an optimal posterior $\pi^*$. As

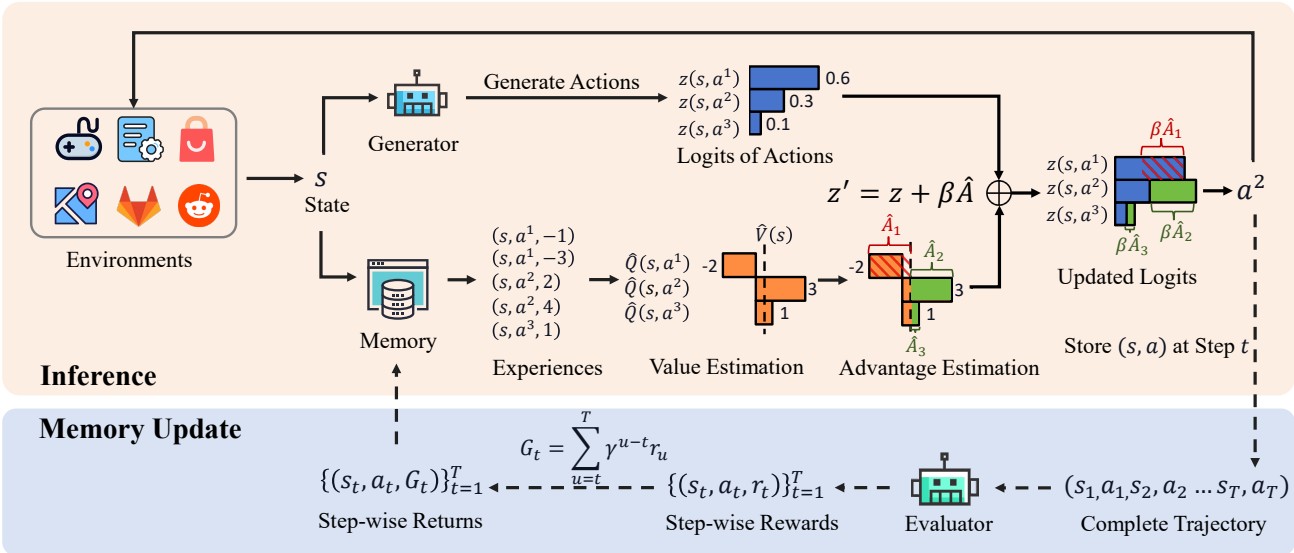

*Figure 2.* **Overview of the Just-In-Time Reinforcement Learning (JitRL) framework.** The system operates in a continuous loop: (1) In the *Inference* (top), the agent retrieves relevant past experiences $\mathcal{N}(s)$ from the non-parametric memory $\mathcal{M}$. The base LLM's logits $z$ are then adjusted in closed-form ($z' = z + \beta\widehat{A}$) using the estimated advantage $A$ derived from historical returns, enabling test-time policy improvement without gradient updates. (2) In the *Memory Update* (bottom), completed trajectories are analyzed by an evaluator to compute discounted returns $G_t$. These new experiences are stored back into $\mathcal{M}$, allowing the agent to evolve its policy across episodes.

illustrated in Figure 2, the framework consists of three key components: (1) constructing a granular experience memory from completed trajectories, (2) estimating state and action values via retrieval during inference, and (3) dynamically updating the LLM's policy logits based on estimated advantages. The algorithm can be found in Appendix A.

### 4.1. Memory Construction

In the absence of a trained value network, we estimate expected returns by querying a dynamic memory $\mathcal{M} = \{(s_i, a_i, G_i)\}_{i=1}^{N}$. Here, $G_i$ characterizes the cumulative discounted reward observed after state $s_i$ and action $a_i$ in prior trajectories, effectively capturing the environment's empirical distribution.

**Reflective Step-wise Rewards.** Credit assignment is challenging under long trajectories, so we use the agent's self-reflective abilities to improve credit assignment. At the end of each episode, an LLM-based **Evaluator** assesses the completed trajectory $\tau = (s_1, a_1, \ldots, s_T, a_T)$ to generate step-wise rewards. This is defined by the mapping $\mathcal{E} : \tau \to \{r_t\}_{t=1}^{T}$, where each $r_t$ is a scalar reward quantifying the individual contribution of action $a_t$ to the overall success of the task. These rewards are then aggregated into the discounted return $G_t$ to quantify the long-term value of each action:

$$G_t = \sum_{u=t}^{T} \gamma^{u-t} r_u, \qquad (3)$$

where $\gamma \in [0, 1]$ balances the weight of immediate versus future rewards.

**State Construction.** Raw states (e.g., full HTML DOM trees or verbose game text) are often too noisy for effective retrieval. We abstract them into compact, structured states $s_t$ that preserve task-relevant semantics while discarding irrelevant details. Our key design principle is mapping functionally equivalent states to similar representations. Details on state construction can be found in Appendix F.

Ultimately, each transition is stored in the dynamic memory $\mathcal{M}$ as a compact triplet $(s_t, a_t, G_t)$, implicitly representing the empirical distribution of the environment's dynamics.

### 4.2. Test-Time Value Estimation

Instead of relying on computationally expensive value networks, we perform on-the-fly value estimation by retrieving relevant transitions from memory. During inference, the current observation is abstracted into a structured state $s$. We then retrieve top-$k$ similar neighbors to form a local neighborhood $\mathcal{N}(s)$.

We formulate the **state value** estimation $\widehat{V}(s)$ for state $s$ as an expectation of the returns across this neighborhood:

$$\widehat{V}(s) := \frac{1}{|\mathcal{N}(s)|} \sum_{i \in \mathcal{N}(s)} G_i. \qquad (4)$$

Similarly, to evaluate a specific candidate action $a$, we con-

dition the **action value** $\widehat{Q}(s,a)$ on the state-action pair $(s,a)$ over the relevant subset $\mathcal{N}(s,a) = \{(s_i, a_i, G_i) \in \mathcal{N}(s) : a_i = a\}$: We distinguish between two cases:

1. **Known Actions:** If historical evidence exists (i.e., $|\mathcal{N}(s,a)| > 0$), we estimate the action value by averaging the returns within the subset:

$$\widehat{Q}(s,a) := \frac{1}{|\mathcal{N}(s,a)|} \sum_{j \in \mathcal{N}(s,a)} G_j. \tag{5}$$

2. **Unseen Actions:** For actions lacking historical data (i.e., $|\mathcal{N}(s,a)| = 0$), we encourage exploration by adopting an optimism under uncertainty principle. With probability $\lambda$, we assign an optimistic value:

$$\widehat{Q}(s,a) := \widehat{V}(s) + \frac{\alpha}{|\mathcal{N}(s)|}. \tag{6}$$

The bonus $\alpha/|\mathcal{N}(s)|$ reflects epistemic uncertainty: sparse memory coverage (small $|\mathcal{N}(s)|$) implies unreliable estimates, warranting exploration. As experiences accumulate, the bonus diminishes, naturally shifting toward exploitation. With probability $1 - \lambda$, we assign $Q(s,a) = 0$ to prevent over-exploration.

The **test-time advantage** is then derived by centering the action value against the state baseline:

$$\widehat{A}(s,a) = \widehat{Q}(s,a) - \widehat{V}(s). \tag{7}$$

This allows us to identify actions that outperform the local average purely through inference-time retrieval, serving as a proxy for the advantage function $A^\pi(s,a)$.

### 4.3. Policy Update

Given these value estimates, we formally derive the policy update as a constrained optimization problem. We seek an optimal policy:

$$\pi^* = \arg\max_{\pi'} \left( \mathbb{E}_{a \sim \pi'}[A(s,a)] - \frac{1}{\beta} D_{KL}(\pi' \| \pi_\theta) \right), \tag{8}$$

where $\beta$ is the temperature hyperparameter. Eq. (8) maximizes the expected advantage while minimizing the KL-divergence from the frozen reference policy $\pi_\theta$ to preserve linguistic coherence. The closed-form solution to this optimization objective is:

$$\pi^*(a|s) \propto \pi_\theta(a|s) \exp\left( \beta A(s,a) \right) \tag{9}$$

where $\beta$ is the temperature parameter that controls the strength of the constraint.

To implement this optimal policy without gradient updates, we map Eq. (9) directly to the logit space. Let $z(s,a)$ be the logits of the base LLM such that $\pi_\theta(a|s) =$

Softmax$(z(s,a))$. Taking the logarithm of Eq. (9) yields an additive update rule:

$$z'(s,a) = z(s,a) + \beta \cdot \widehat{A}(s,a) \tag{10}$$

By applying the Softmax function to the updated logits $z'(s,a)$, we recover the optimal policy distribution $\pi^*$.

### 4.4. Theoretical Analysis

We provide theoretical justification for JitRL through three progressive steps. First, we prove our logit update is the exact closed-form solution for KL-constrained optimization (Theorem 4.1). Next, we demonstrate that our value estimates converge to the true values (Theorem 4.2), ensuring the overall policy update consistently converges to the optimal policy (Theorem 4.3). Detailed statements and proofs can be found in Appendix B, C, and D.

**Optimality of Logit Update.** We formulate the inference-time adjustment as a constrained optimization problem. Our goal is to find a policy $\pi'$ that effectively utilizes the retrieved advantage information while preserving the linguistic capabilities of the pre-trained model.

**Theorem 4.1** (Optimality of Policy Update). *Let $\pi_\theta$ be the reference policy with logits $z(s,a)$. Consider the problem of finding the optimal policy $\pi^*$ that maximizes the expected advantage subject to a KL-divergence penalty:*

$$\pi^* = \arg\max_{\pi'} \left( \mathbb{E}_{a \sim \pi'}[\widehat{A}(s,a)] - \frac{1}{\beta} D_{KL}(\pi' \| \pi_\theta) \right) \tag{11}$$

*where $\beta$ is a temperature hyperparameter. The solution to this optimization problem is given by the additive logit update:*

$$z'(s,a) = z(s,a) + \beta \cdot \widehat{A}(s,a) \tag{12}$$

*where $z'(s,a)$ denotes the logits of the optimal policy $\pi^*$.*

**Consistency of Value and Advantage Estimates.** Next, we show that the estimators $\widehat{V}_t$, $\widehat{Q}_t$, and $\widehat{A}_t$ converge in probability to their true counterparts $V^{\pi_t}$, $Q^{\pi_t}$, and $A^{\pi_t}$, even under the non-stationary policy setting $(\pi_t)_{t \geq 1}$.

**Theorem 4.2** (Consistency of $\widehat{V}_t$, $\widehat{Q}_t$, and $\widehat{A}_t$). *Under the assumptions given in Appendix C, for any fixed query state $s$ and action $a$ at time $t$, as $t \to \infty$,*

$$\widehat{V}_t(s) \xrightarrow{p} V^{\pi_t}(s),$$
$$\widehat{Q}_t(s,a) \xrightarrow{p} Q^{\pi_t}(s,a),$$
$$\widehat{A}_t(s,a) \xrightarrow{p} A^{\pi_t}(s,a).$$

**Consistency of Policy Update.** Finally, combining the previous two theorems, we conclude that our policy update converges to the KL-regularized optimal policy induced by the true advantages $A^{\pi_t}$:

**Theorem 4.3** (Consistency of Policy Update). *Fix a state s, and define*

$$\pi_t^*(a \mid s) \propto \pi_\theta(a \mid s) \exp\big(\beta A^{\pi_t}(s, a)\big),$$
$$\widehat{\pi}_t(a \mid s) \propto \pi_\theta(a \mid s) \exp\big(\beta \widehat{A}_t(s, a)\big).$$

*Under the assumptions of Theorem 4.2, as $t \to \infty$,*

$$\widehat{\pi}_t(\cdot \mid s) \xrightarrow{p} \pi_t^*(\cdot \mid s),$$

*for any finite candidate action set (or more generally, under uniform convergence of $\widehat{A}_t(s, \cdot)$ on the candidate set).*

## 5. Experiments

In this section, we design our experiments to answer the following four research questions:

- **RQ1:** How does JitRL compare against state-of-the-art training-free baselines and weight-update methods?
- **RQ2:** Does JitRL possess generalization capabilities across unseen tasks and different model backbones?
- **RQ3:** How does JitRL qualitatively correct the agent's decision-making?
- **RQ4:** How do key components impact JitRL's performance?

### 5.1. Experimental Setup

**Benchmarks.** We evaluate our method in two distinct environments to demonstrate versatility:

- **WebArena** (Zhou et al., 2024): A realistic web environment comprising multiple functional websites. It challenges agents to navigate complex DOM trees and execute sequential actions to fulfill user instructions.
- **Jericho** (Hausknecht et al., 2020): A benchmark suite for interactive fiction games where agents interact purely via textual commands. We evaluate performance on three representative games: Library, Zork 1, and Zork 3.

**Baselines.** We compare JitRL against a comprehensive set of methods, which fall into two paradigms: training-free and weight-update.

- **Training-Free Methods.** We evaluate five inference-time strategies: (1) **Static**: A non-learning agent with fixed configuration. (2) **Memory**: An in-context learning baseline that fills the context window with full transcripts of past episodes, utilizing a First-In-First-Out (FIFO) strategy to handle token limits. (3) **Reflexion** (Shinn et al., 2023): A prompt-based method where the agent generates structured textual self-reflections after each episode to guide subsequent attempts. (4) **AWM** (Wang et al., 2025): An episodic memory approach that extracts and persistently stores reusable workflows derived from successful trajectories. (5) **EvoTest** (He et al., 2025): An evolutionary

optimization framework that iteratively rewrites prompts, updates memory with state-action pairs, and dynamically tunes hyperparameters. For JitRL and training-free baselines, we use `Gemini-2.5-flash` as LLM backbones.

- **Weight-Update Methods**. We compare against leading training approaches. For WebArena, we directly utilize the pre-trained checkpoints for **SFT** and **WebRL** provided by (Qi et al., 2025), both built on `Llama-3.1-70B-Instruct`. For Jericho, we employ the task-specific `Qwen3-32B` checkpoints, which are trained via **GRPO** (Shao et al., 2024) for each individual game. Refer to Appendix E for a detailed description of the training process.

**Implementation Details.** We implement two variants to derive logits for the logit-based policy update. This design specifically addresses the constraint that many black-box models do not expose access to raw log-probabilities:

- **Token-level Logit**: We prompt the generator to select an action from $k$ candidates by outputting a corresponding index token (e.g., "1", "2"). We then extract the log-probabilities of these tokens as logits.
- **Verbalized Logit**: For models that do not expose log-probabilities, we prompt the LLM to explicitly output a confidence score (0–100) and transform them to logits for each candidate action.

More implementation details, such as state representation, memory retrieval, and action handling, can be found in Appendix F. Prompt templates can be found in Appendix G, hyperparameters can be found in Appendix H.

### 5.2. RQ1: Main Performance Comparison

To address RQ1, we employ a multi-trial sequential testing protocol. Instead of evaluating each task only once, the agent executes $L$ consecutive episodes for each task $i$. This setup is designed to simulate a realistic deployment scenario where an agent must improve its performance on-the-fly through repeated interactions.

#### 5.2.1. RESULTS ON WEBARENA

**Metrics.** Our evaluation dataset contains $\mathcal{T}$ tasks, which we run sequentially, each for $L = 5$ times, and evaluate performance primarily using the Success Rate. Let $y_{i,j} \in \{0, 1\}$ denote the binary completion status (success=1, failure=0) of task $i$ at episode $j$. We report two aggregated metrics: (1) **Average Success Rate (Avg)**, calculated as $\frac{1}{\mathcal{T} \cdot 5} \sum_{i=1}^{\mathcal{T}} \sum_{j=1}^{5} y_{i,j}$, which averages performance across all attempts to reflect the overall learning efficiency; and (2) **Final Success Rate (Final)**, computed as $\frac{1}{\mathcal{T}} \sum_{i=1}^{\mathcal{T}} y_{i,5}$, which measures the success rate of the final episode to indicate the agent's converged capability. Notably, a larger differential between Final and Avg indicates

*Table 1.* Main results on WebArena. We report average (**Avg**) and final (**Final**) success rate (%) for each domain. The gap between Final and Avg reflects learning efficiency. The final column (Average) is calculated as the micro-average across all tasks from all websites.

| Method | Admin | | GitLab | | Map | | Reddit | | Shopping | | Average | |
|---|---|---|---|---|---|---|---|---|---|---|---|---|
| | Avg | Final | Avg | Final | Avg | Final | Avg | Final | Avg | Final | Avg | Final |
| Static | 39.46 | 39.67 | 38.92 | 38.24 | 30.62 | 31.25 | 39.60 | 42.86 | 24.06 | 25.00 | 35.63 | 36.30 |
| Memory | 47.91 | 47.80 | 39.31 | 40.20 | 30.47 | 32.81 | 53.02 | 55.04 | 30.16 | 33.16 | 41.36 | 43.00 |
| Reflexion | 48.46 | 50.55 | 38.43 | 40.69 | 29.38 | 27.34 | 54.88 | 56.59 | 30.83 | 31.25 | 41.08 | 42.12 |
| AWM | 49.46 | 51.09 | 37.94 | 40.20 | 34.38 | 32.81 | 55.66 | 58.14 | 23.12 | 22.40 | 39.37 | 40.32 |
| EvoTest | 44.51 | 47.80 | 37.94 | 39.22 | 33.59 | 41.41 | 51.78 | 54.26 | 26.67 | 29.17 | 39.24 | 42.49 |
| **JitRL** | **52.31** | **56.59** | **40.78** | **45.00** | **37.66** | **42.19** | **57.64** | **61.98** | **41.67** | **45.83** | **46.98** | **51.35** |

*Table 2.* Comparison of **Final** success rate (%) with weight-update methods on **WebArena-Lite**, the held-out test set. The remainder of WebArena was used to train WebRL.

| Method | Admin | GitLab | Map | Reddit | Shopping | Average |
|---|---|---|---|---|---|---|
| SFT | 20.00 | 20.00 | 26.70 | 52.60 | 13.30 | 23.00 |
| WebRL | 58.33 | 47.06 | 32.26 | 62.50 | 30.43 | 46.06 |
| **JitRL** | **65.71** | **56.67** | **53.57** | **78.95** | **53.33** | **60.00** |

*Table 3.* Results on Jericho games. We report: **Avg**: average score across 50 episodes; **Final**: final episode score.

| Method | Library | | Zork1 | | Zork3 | |
|---|---|---|---|---|---|---|
| | Avg | Final | Avg | Final | Avg | Final |
| Static | 10.0 | 10 | 8.5 | 10 | 0.2 | 0 |
| Memory | 13.2 | 14 | 22.9 | 25 | 1.0 | 1 |
| Reflexion | 15.4 | 18 | 26.1 | 35 | 1.4 | 1 |
| AWM | 12.7 | 10 | 38.8 | 44 | 1.9 | 2 |
| EvoTest | 21.5 | 26 | 46.8 | 54 | 2.6 | 4 |
| GRPO | 13.6 | 11 | 16.2 | 10 | 1.1 | 2 |
| **JitRL** | **25.9** | **30** | **53.0** | **69** | **3.1** | **5** |

a steeper learning curve, demonstrating that the agent successfully leverages historical memory to improve its policy over time.

Table 1 presents the performance comparison of JitRL and training-free baselines on WebArena. JitRL outperforms all training-free baselines in both cumulative (Avg) and converged (Final) success rates. Moreover, the significant gap between the Avg and Final highlights JitRL's robust learning capability. Performance gains are most pronounced in structured domains like Shopping (+73.2% over Static) due to high trajectory reusability. In contrast, Reflexion occasionally suffers from "reflection noise" in sites like Map, where misleading feedback degrades performance.

Table 2 compares JitRL with weight-update methods (WebRL) on the held-out WebArena-Lite (Qi et al., 2025) subset, as the remaining WebArena tasks were utilized for training WebRL and SFT. JitRL attains SOTA performance purely through inference-time optimization. This demonstrates that JitRL serves as a highly efficient alternative to computation-ally intensive training methods. We also provide results with the same backbone in an on-the-fly setting (Appendix I), and with the same backbone following separate training and evaluation phases (Appendix J.1).

### 5.2.2. RESULTS ON JERICHO

**Metrics.** For Jericho, we evaluate the **Game Score**. Let $y_{i,j}$ represent the score obtained in game $i$ during episode $j$. Similar to WebArena, we report the **Average Score (Avg)** over 50 episodes to capture the learning stability, and the **Final Score (Final)** to assess the performance at the final episode.

Table 3 presents the quantitative comparison on Jericho. JitRL achieves the highest scores across all three games, significantly outperforming baselines. The learning curves for each episode are visualized in Figure 3, which reveal several key insights: (1) **JitRL exhibits rapid initial learning**, achieving competitive performance within the first 10-15 episodes; (2) **the performance gap between JitRL and baselines widens as episodes progress**, indicating effective experience accumulation; and (3) **JitRL shows reduced variance in later stages**, implying stable policy convergence. In contrast, GRPO exhibits high variance throughout, suggesting that gradient-based updates struggle with the sparse reward signals. Memory-based approaches (Memory, AWM) plateau early, because they over-rely on established memory patterns, which inhibit further exploration and cause the agents to converge to suboptimal policies. A same-backbone comparison between JitRL and GRPO is provided in Appendix J.2.

### 5.3. RQ2: Generalization Capabilities

To answer RQ2, we assess the universality of our framework. This section evaluates JitRL's performance across different LLM backbones and investigates its zero-shot generalization on unseen tasks where no direct historical memory exists.

**Generalization Across Backbones.** We further evaluate JitRL on GPT-5-mini and DeepSeek-V3.2 (Sands et al.,

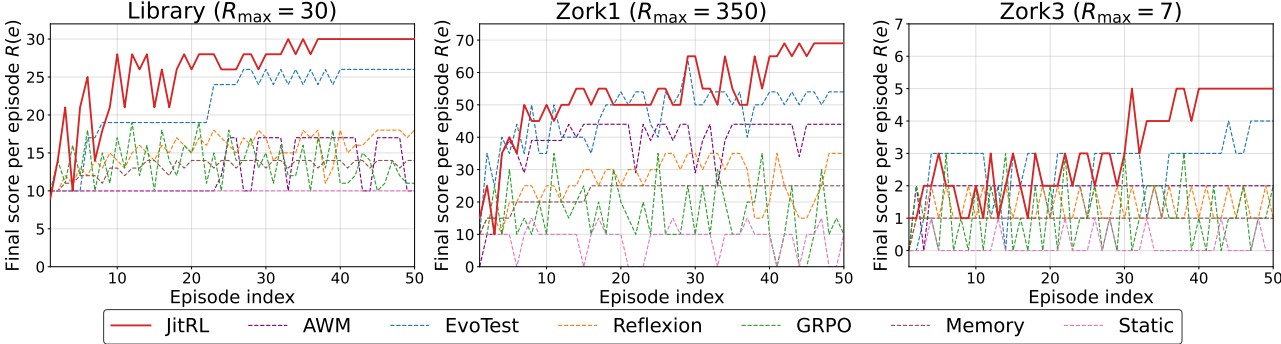

*Figure 3.* Learning curves on Jericho games. JitRL shows consistent improvement across episodes.

*Table 4.* Comparison of average (**Avg**) and **Final** success rate between JitRL and baseline methods across different backbone LLMs.

| Method | Admin | | Reddit | |
|---|---|---|---|---|
| | Avg | Final | Avg | Final |
| *Gemini-2.5-flash* | | | | |
| Static | 39.46 | 39.67 | 39.60 | 42.86 |
| Memory | 47.91 | 47.80 | 53.02 | 55.04 |
| Reflexion | 48.46 | 50.55 | 54.88 | 56.59 |
| AWM | 49.46 | 51.09 | 55.66 | 58.14 |
| EvoTest | 44.51 | 47.80 | 51.78 | 54.26 |
| **JitRL (Ours)** | **52.31** | **56.59** | **57.64** | **61.98** |
| *GPT-5-mini* | | | | |
| Static | 37.07 | 37.50 | 47.29 | 46.51 |
| Memory | 43.04 | 45.65 | 50.54 | 51.94 |
| Reflexion | 40.43 | 41.30 | 49.15 | 49.61 |
| AWM | 40.00 | 40.22 | 50.23 | 51.94 |
| EvoTest | 42.93 | 43.48 | 51.94 | 53.49 |
| **JitRL (Ours)** | **48.04** | **51.63** | **54.26** | **57.36** |
| *DeepSeek-V3.2* | | | | |
| Static | 32.07 | 32.61 | 42.02 | 43.41 |
| Memory | 37.07 | 45.65 | 51.47 | 56.28 |
| Reflexion | 38.04 | 40.22 | **55.04** | 56.59 |
| AWM | 42.07 | 44.02 | 52.71 | 55.04 |
| EvoTest | 44.35 | 45.65 | 50.85 | 53.49 |
| **JitRL (Ours)** | **50.65** | **54.35** | 54.42 | **61.24** |

*Table 5.* Generalization to Unseen Tasks. We report the average success rate (%) where the agent is restricted to retrieving memories exclusively from disjoint tasks.

| Method | Admin | GitLab | Map | Reddit | Shopping |
|---|---|---|---|---|---|
| Static | 36.96 | 37.25 | 31.19 | 36.43 | 23.44 |
| Memory | 47.83 | 37.75 | 33.03 | 50.39 | 26.04 |
| Reflexion | 46.74 | 34.80 | 31.19 | 48.84 | 31.77 |
| AWM | 47.28 | 36.27 | 34.86 | 53.49 | 23.44 |
| EvoTest | 43.48 | 35.78 | 34.86 | 49.61 | 18.75 |
| **JitRL** | **48.37** | **38.73** | **35.78** | **55.04** | **36.98** |

*Table 6.* Distribution (%) of cross-task memory utilization.

| Admin | GitLab | Map | Reddit | Shopping | Avg |
|---|---|---|---|---|---|
| 40.54 | 38.47 | 37.64 | 56.31 | 62.19 | 47.03 |

### 5.4. Qualitative Analysis: Correcting Semantic Priors

To provide an intuitive understanding of JitRL's mechanism, we examine representative cases where the retrieved memory successfully overrides the base model's erroneous intuition. As shown in Table 7, regarding **Site Functionality**, the agent learns to ignore intuitive but incorrect links (e.g., Catalog") and instead navigates to the correct section (Marketing"). Similarly, for **Navigation Precision**, memory steers the agent away from noisy global searches toward deterministic navigation links. Additionally, in terms of **UI Mechanics**, JitRL optimizes interaction efficiency, identifying shortcuts such as prioritizing hover actions to directly reveal fine-grained subcategories rather than clicking into broad category pages. Additional case studies on Jericho can be found in Appendix K.

### 5.5. Qualitative Analysis: Importance of Cross-Task Memory

We observe the retrieval patterns when both cross-task memory and cross-episode memory are enabled. As shown in Table 6, cross-task memory accounts for nearly 50% of

2025) to assess its generalizability against other baselines. As shown in Table 4, JitRL achieves state-of-the-art performance in most cases. This indicates that our logit-update mechanism is model-agnostic and robustly enhances the capabilities of efficient models without parameter tuning.

**Cross Task Generalization.** We assess cross-task generalization on WebArena by simulating a cold-start setting, where the agent utilizes memories solely from disjoint tasks. As shown in Table 5, JitRL consistently outperforms baselines, indicating effective transfer of abstract procedural knowledge rather than specific solutions.

*Table 7.* **Qualitative Analysis of Policy Improvement.** We select three representative cases showing how JitRL modifies decision-making. **Base** and **JitRL** denote the logits before and after the memory-based update. Compared to **Base**, **JitRL** decreases the logits for **incorrect** options and increases them for **correct** options. In this way, JitRL corrects semantic priors and optimizes efficiency.

| Task | Candidate Action | Base | JitRL | Mechanism Explanation |
|---|---|---|---|---|
| Find customer reviews | **click(CATALOG)** | **0.90** | 0.40 | While "Catalog" appears semantically intuitive, memory |
| (Site Functionality) | **click(MARKETING)** | 0.70 | **1.40** | corrects this prior: reviews are located under "Marketing". |
| Find posts in subreddit | **fill(Search, "..")** | **0.95** | 0.45 | Global search often yields noisy results. Memory steers the |
| (Navigation Precision) | **click(Forums)** | 0.80 | **1.80** | agent toward "Forums" for a deterministic and accurate path. |
| Access product inventory | **click(Products)** | **0.95** | -0.28 | Clicking loads a generic page. Memory identifies "hover" as |
| (UI Mechanics) | **hover(Products)** | 0.40 | **0.90** | an efficient shortcut to instantly reveal subcategories. |

*Table 8.* Ablation: Logit update vs. prompting on WebArena.

| Method | Admin | Reddit |
|---|---|---|
| JitRL (Prompt Update) | 49.46 | 53.02 |
| JitRL (Logit Update) | **52.31** | **57.64** |

the retrieved context, indicating a substantial reliance on this mechanism alongside cross-episode memory. Furthermore, websites that trigger a higher frequency of cross-task memory retrieval also demonstrate superior cross-task generalization performance compared to Static.

### 5.6. RQ4: Ablation Studies

Finally, we address RQ4 by analyzing the sensitivity of JitRL to key components.

**Impact of Logit Update.** To investigate whether the performance gain stems from the logit update mechanism or just the retrieved information, we compare JitRL's **Logit Update** against **Prompt Update**, which uses the exact same memory retrieval but appends them to the prompt instead of modifying logits. Table 8 shows that direct Logit Update outperforms Prompt Update on Admin and Reddit websites. We attribute this to the limitations of prompting: as context length grows, LLMs often struggle to attend to retrieved cues or fail to strictly follow instructions. In contrast, our logit update directly modulates the output distribution, ensuring the agent effectively exploits historical advantages.

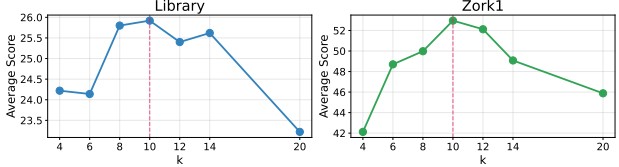

*Figure 4.* Impact of Retrieval Neighbor Count ($k$).

**Impact of Retrieval Neighbor Count.** We evaluate the impact of retrieval neighbor count ($k$) on Library and Zork1.

As shown in Figure 4, the agent achieves robust performance with $k$ ranging from 8 to 14. Outside this range, performance degrades due to the trade-off between context sufficiency and noise: a small $k$ limits the historical evidence needed for stable value estimation, leading to high variance, while a large $k$ introduces excessive noise that hinders efficient exploration and delays convergence.

More ablation studies such as the impact of state representation can be found in Appendix L.

### 5.7. Cost Comparison

We compare the monetary costs of JitRL against all baselines. As shown in Table 9, the weight-update method, WebRL, incurs massive overheads when trained on `Llama-3.1-70B-Instruct`. The costs are estimated based on training expenses on NVIDIA H200 GPUs. In contrast, JitRL aligns with other training-free methods whose costs are calculated based on API usage prices. Despite its lower cost, JitRL achieves superior performance, as demonstrated in our main experiments. The detailed cost analysis can be found in Appendix R.

*Table 9.* Comparison of training cost.

| Metric | Static | Memory | Reflexion | AWM | EvoTest | WebRL | JitRL |
|---|---|---|---|---|---|---|---|
| **Cost** | $200 | $230 | $220 | $250 | $220 | ~$9900 | $290 |

## 6. Conclusion

In this work, we propose Just-In-Time Reinforcement Learning (JitRL), a training-free framework that enables frozen LLMs to continuously adapt at test time by directly optimizing logits. We theoretically prove that our update rule is the optimal closed-form solution for RL policy optimization. Experimental results show that JitRL not only outperforms existing training-free methods but also outperforms expensive weight-update baselines, while reducing monetary costs by over 30×, offering a scalable and efficient path for agentic continuous learning.

## Limitations

Despite its strong empirical performance, JitRL has several limitations that point to directions for future work.

**Dependence on the frozen base model.** JitRL re-weights actions over the candidate set proposed by the base model; it cannot discover actions that the base model would never generate. Furthermore, JitRL relies on an LLM evaluator to produce accurate step-wise rewards — if the evaluator misattributes credit, the resulting advantage estimates will be inaccurate and may degrade policy quality.

**Less suited for tasks where key information is hard to represent in text.** JitRL's state representation and retrieval operate on text, so for tasks where critical patterns are difficult to express textually — such as spatial reasoning (e.g., chess board positions) or time-series forecasting — text-based retrieval may fail to capture the relevant similarity between states, limiting the effectiveness of memory-based value estimation.

## Impact Statement

By eliminating gradient-based updates, JitRL reduces the cost of continual agent learning by over $30\times$, making adaptive agents accessible without large-scale GPU infrastructure. However, the memory bank stores real interaction trajectories, which may inadvertently capture sensitive user information in real-world deployments.

## Acknowledgement

This research is supported by the Ministry of Education, Singapore, under the Academic Research Fund Tier 2 (FY2025) (Grant T2EP20124-0038). Computational work involved in this research work is partially supported by NUS IT's Research Computing group under grant number NUSREC-HPC-00001.

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

## A. Algorithm

Algorithm 1 shows the overall algorithm of JitRL.

## B. Derivation of the Optimal Closed-Form Solution

In this appendix, we provide the full derivation for Theorem 4.1. We show that the additive logit update rule is the exact solution to the KL-constrained advantage maximization problem.

### B.1. Problem Formulation

Recall our optimization objective defined in Eq. (8). We seek a policy distribution $\pi'$ that maximizes the expected estimated advantage while maintaining a bounded KL divergence from the reference policy $\pi_\theta$:

$$\max_{\pi'} \quad \mathbb{E}_{a \sim \pi'}[\widehat{A}(s, a)] - \frac{1}{\beta} D_{KL}(\pi' || \pi_\theta). \tag{13}$$

Subject to the probability constraint that the distribution must sum to 1:

$$\sum_a \pi'(a) = 1. \tag{14}$$

Expanding the expectation and KL divergence terms, the objective function $J(\pi')$ can be written as:

$$J(\pi') = \sum_a \pi'(a)\widehat{A}(s, a) - \frac{1}{\beta} \sum_a \pi'(a) \log \frac{\pi'(a)}{\pi_\theta(a)}. \tag{15}$$

### B.2. Lagrangian Optimization

To solve this constrained optimization problem, we introduce a Lagrange multiplier $\lambda$ for the summation constraint. The Lagrangian function $\mathcal{L}(\pi', \lambda)$ is:

$$\mathcal{L}(\pi', \lambda) = \sum_a \pi'(a)\widehat{A}(s, a) - \frac{1}{\beta} \sum_a \pi'(a)(\log \pi'(a) - \log \pi_\theta(a)) + \lambda \left( \sum_a \pi'(a) - 1 \right). \tag{16}$$

We take the derivative of $\mathcal{L}$ with respect to $\pi'(a)$ for a specific action $a$ and set it to zero:

$$\frac{\partial \mathcal{L}}{\partial \pi'(a)} = \widehat{A}(s, a) - \frac{1}{\beta}(1 + \log \pi'(a) - \log \pi_\theta(a)) + \lambda = 0. \tag{17}$$

Rearranging the terms to solve for $\log \pi'(a)$:

$$\frac{1}{\beta} \log \pi'(a) = \widehat{A}(s, a) + \frac{1}{\beta} \log \pi_\theta(a) + \lambda - \frac{1}{\beta}. \tag{18}$$

Multiplying by $\beta$:

$$\log \pi'(a) = \beta\widehat{A}(s, a) + \log \pi_\theta(a) + (\beta\lambda - 1). \tag{19}$$

Exponentiating both sides gives the form of the optimal policy:

$$\pi'(a) = \pi_\theta(a) \exp(\beta\widehat{A}(s, a)) \cdot \exp(\beta\lambda - 1). \tag{20}$$

Here, the term $\exp(\beta\lambda - 1)$ is constant across all actions $a$ and serves as the normalization factor (partition function) $Z$ to ensure $\sum \pi'(a) = 1$. Thus, we can write:

$$\pi'(a) = \frac{1}{Z}\pi_\theta(a)\exp(\beta\widehat{A}(s,a)) \propto \pi_\theta(a)\exp(\beta\widehat{A}(s,a)). \tag{21}$$

is the exact implementation of the optimal policy update in logit space. This completes the proof. $\qquad\square$

## C. Consistency of Values and Advantages Under Nonstationary Policies

### C.1. Setup and Assumptions

We consider the nonstationary setting in which memory is generated under a sequence of policies $(\pi_t)_{t\geq 1}$. Each memory entry $i$ is a triplet $(S_i, A_i, G_i)$ collected at time $t(i)$ under policy $\pi_{t(i)}$. For a query at time $t$, estimators are formed using the current memory of size $N = N(t)$.

**Assumption C.1** (State regularity). For all policies $\pi$:

$$|V^\pi(s) - V^\pi(s')| \leq L_V\, d(s, s'),$$
$$|Q^\pi(s, a) - Q^\pi(s', a)| \leq L_Q\, d(s, s'),$$

for all states $s, s'$ and actions $a$.

**Assumption C.2** (Noise model). For each memory entry,

$$G_i = Q^{\pi_{t(i)}}(S_i, A_i) + \epsilon_i, \qquad \mathbb{E}[\epsilon_i \mid S_i, A_i] = 0, \qquad \mathrm{Var}(\epsilon_i \mid S_i, A_i) \leq \sigma^2.$$

**Assumption C.3** (kNN regime and coverage). As $t \to \infty$, the memory size $N(t) \to \infty$, and the neighborhood size $k = k(N)$ satisfies

$$k \to \infty, \qquad k/N \to 0.$$

The state distribution has support covering the query region.

**Assumption C.4** (Action frequency). For any fixed $(s, a)$, the action-filtered count

$$k_a := |N(s, a)|$$

, satisfies $k_a \to \infty$ in probability.

**Assumption C.5** (Slow policy drift). Define the neighborhood policy drift at time $t$:

$$\Delta_t := \max_{i \in N(s)} \sup_{s'} \mathrm{TV}\big(\pi_{t(i)}(\cdot \mid s'),\, \pi_t(\cdot \mid s')\big).$$

Assume $\Delta_t \to 0$ as $t \to \infty$.

**Assumption C.6** (Policy regularity). For any policies $\pi, \pi'$,

$$|Q^\pi(s, a) - Q^{\pi'}(s, a)| \leq L_\pi \sup_{s'} \mathrm{TV}(\pi(\cdot \mid s'), \pi'(\cdot \mid s')),$$
$$|V^\pi(s) - V^{\pi'}(s)| \leq L_\pi \sup_{s'} \mathrm{TV}(\pi(\cdot \mid s'), \pi'(\cdot \mid s')).$$

**Theorem C.7** (Tracking consistency of $\widehat{V}_t$, $\widehat{Q}_t$, and $\widehat{A}_t$). *Under the above assumptions, for any fixed query state $s$ and action $a$ at time $t$,*
$$\widehat{V}_t(s) \xrightarrow{p} V^{\pi_t}(s), \qquad \widehat{Q}_t(s, a) \xrightarrow{p} Q^{\pi_t}(s, a), \qquad \widehat{A}_t(s, a) \xrightarrow{p} A^{\pi_t}(s, a),$$

*as $t \to \infty$.*

*Proof.* Recall

$$\widehat{Q}_t(s, a) := \frac{1}{k_a} \sum_{i \in N(s,a)} G_i.$$

Substituting the noise model,

$$\widehat{Q}_t(s,a) - Q^{\pi_t}(s,a) = \frac{1}{k_a} \sum_{i \in N(s,a)} \left(Q^{\pi_{t(i)}}(S_i,a) - Q^{\pi_t}(s,a)\right) + \frac{1}{k_a} \sum_{i \in N(s,a)} \epsilon_i.$$

We decompose the first term as

$$Q^{\pi_{t(i)}}(S_i,a) - Q^{\pi_t}(s,a) = \underbrace{Q^{\pi_{t(i)}}(S_i,a) - Q^{\pi_{t(i)}}(s,a)}_{\text{state mismatch}} + \underbrace{Q^{\pi_{t(i)}}(s,a) - Q^{\pi_t}(s,a)}_{\text{policy drift}}.$$

**State mismatch.** By state regularity,

$$|Q^{\pi_{t(i)}}(S_i,a) - Q^{\pi_{t(i)}}(s,a)| \leq L_Q \, d(S_i,s).$$

Under the kNN regime, the average neighbor distance converges to zero in probability, so this term vanishes.

**Policy drift.** By policy regularity,

$$|Q^{\pi_{t(i)}}(s,a) - Q^{\pi_t}(s,a)| \leq L_\pi \Delta_t \to 0.$$

**Variance.** Conditioned on $\{(S_i, A_i)\}$, the noise average has mean zero and variance at most $\sigma^2/k_a$, which converges to zero since $k_a \to \infty$.

Combining the three terms yields $\widehat{Q}_t(s,a) \xrightarrow{p} Q^{\pi_t}(s,a)$. The result for $\widehat{V}_t$ follows identically, and $\widehat{A}_t = \widehat{Q}_t - \widehat{V}_t$ converges as the difference of convergent estimators is convergent, by Slutsky's theorem. □

## D. Consistency of Policy Update

Finally, combining the previous theorems, we conclude that our policy update converges to the KL regularized policy defined by the true advantage values.

**Theorem D.1** (Consistency of the closed-form policy update). *Fix a state $s$. Define*

$$\pi_t^*(a \mid s) \propto \pi_\theta(a \mid s) \exp\left(\beta A^{\pi_t}(s,a)\right),$$
$$\widehat{\pi}_t(a \mid s) \propto \pi_\theta(a \mid s) \exp\left(\beta \widehat{A}_t(s,a)\right).$$

*Under the assumptions of Theorem 4.2, as $t \to \infty$,*

$$\widehat{\pi}_t(\cdot \mid s) \xrightarrow{p} \pi_t^*(\cdot \mid s),$$

*for any finite candidate action set (or more generally, under uniform convergence of $\widehat{A}_t(s,\cdot)$ on the candidate set).*

*Proof.* For a fixed state $s$, define the mapping $\Phi : \mathbb{R}^{|\mathcal{A}|} \to \Delta(\mathcal{A})$ by

$$\Phi(x)(a) = \frac{\pi_\theta(a \mid s) \exp(\beta x(a))}{\sum_{a' \in \mathcal{A}} \pi_\theta(a' \mid s) \exp(\beta x(a'))}.$$

When $\mathcal{A}$ is finite, $\Phi$ is continuous everywhere since it is a composition of continuous functions and the denominator is strictly positive. By Theorem 4.2, the vector $\widehat{A}_t(s,\cdot)$ converges in probability to $A^{\pi_t}(s,\cdot)$ componentwise, hence also as a vector in $\mathbb{R}^{|\mathcal{A}|}$. Therefore, by the continuous mapping theorem,

$$\widehat{\pi}_t(\cdot \mid s) = \Phi\left(\widehat{A}_t(s,\cdot)\right) \xrightarrow{p} \Phi\left(A^{\pi_t}(s,\cdot)\right) = \pi_t^*(\cdot \mid s),$$

which proves the claim. □

# E. Baselines

**Static.** As a non-adaptive reference, this agent utilizes a fixed configuration throughout the entire evaluation. It relies on a consistent, manually-engineered prompt and predefined hyperparameters, maintaining a frozen state with no inter-episode updates or memory retention. Its primary role is to establish a performance floor, representing the backbone LLM's zero-shot capability. Consequently, any performance fluctuations are purely a byproduct of the model's inherent decoding stochasticity.

**Memory.** This approach leverages In-Context Learning (ICL) by providing the model with the complete historical record of the session. Following each episode, the entire interaction transcript is concatenated to a persistent log, which is then fed into the model's context window for the next iteration. While this allows the agent to potentially induce strategies from past experiences, it is fundamentally constrained by the LLM's context length; once this limit is reached, older information is discarded via a first-in-first-out (FIFO) truncation mechanism.

**Reflexion.** (Shinn et al., 2023) Representing a linguistic paradigm of reinforcement learning, this method introduces an explicit self-correction loop. At the end of each episode, the agent critiques its own trajectory to diagnose failures and propose concrete behavioral adjustments (e.g., identifying a missing prerequisite like a 'key' before attempting to 'open a door'). These insights are distilled into a concise textual summary that is prepended to the subsequent prompt, thereby accumulating a strategic memory that guides future reasoning.

**Evotest.** (He et al., 2025) An evolutionary test-time learning framework designed to optimize the entire agentic system without gradient-based fine-tuning. Unlike previous methods that focus on isolated memory or textual feedback, EvoTest employs a dual-role architecture: an *Actor Agent* for task execution and an *Evolver Agent* for system-level refinement. After each episode, the Evolver analyzes the full transcript to iteratively evolve the agent's configuration for the subsequent run. This evolution encompasses a multi-faceted optimization: it systematically rewrites the system prompt, updates a persistent memory of effective state-action pairs, dynamically tunes hyperparameters, and refines tool-use routines. By treating the agent's entire operational logic as an evolvable artifact, EvoTest can capture complex cross-episode strategies that remain inaccessible to simple reflection or raw memory concatenation.

**AWM.** (Wang et al., 2025) A framework designed to bridge the gap in long-horizon task execution by inducing and reusing structured task workflows. Unlike raw transcript-based memory, AWM focuses on extracting common routines—abstracted sequences of successful actions—from prior experiences. These workflows serve as high-level procedural guidance for the agent in subsequent episodes. AWM is capable of operating in both offline and online modes: in the latter, the agent dynamically induces workflows from test-time queries on the fly. By filtering and providing only the most relevant workflows for a given task, AWM enables the agent to navigate complex action trajectories more efficiently, significantly reducing the number of steps required for success and improving robustness across domain shifts.

For Jericho games, the training details of GRPO are:

**GRPO (Online).** (Shao et al., 2024) This baseline implements a gradient-centric Reinforcement Learning (RL) strategy to iteratively refine the agent's policy. Unlike methods that rely on static prompts, GRPO utilizes state-action trajectories and their corresponding rewards to derive policy gradients after each episode. By updating the underlying model weights, it strengthens the probability of high-reward behaviors while penalizing suboptimal choices. While this approach provides superior credit assignment compared to Supervised Fine-Tuning (SFT), its performance is contingent upon the availability of dense and high-fidelity reward signals. Furthermore, the reliance on backpropagation makes it computationally demanding, requiring substantial GPU overhead for online parameter updates.

Regarding the **WebArena** benchmark, for the **SFT** baseline, we report the original results documented by Qi et al. (2025) as their model checkpoints were not publicly released. For **WebRL**, however, we conducted an independent evaluation by re-running the official checkpoints provided by the authors to ensure a consistent comparison within our experimental environment.

**SFT.** The base models are optimized using a dataset of approximately 1k human-labeled expert demonstrations. This process employs a standard imitation learning objective to maximize the likelihood of expert actions given the corresponding web observations. The resulting SFT model serves as the foundation for subsequent stages, specifically providing the initial weights for the Actor policy and the backbone for the Critic network, the latter of which is augmented with a randomly initialized value head.

**WebRL.** (Qi et al., 2025) This method is a self-evolving online curriculum reinforcement learning framework designed to address the challenges of task scarcity and sparse feedback in web environments. The training process is centered around

a self-evolving curriculum where an evolver agent dynamically generates new training instructions based on previous unsuccessful attempts, which are then filtered by a critic to ensure the tasks match the agent's current proficiency. To provide feedback in the absence of environmental signals, a trained Outcome-supervised Reward Model (ORM) evaluates the final state of trajectories to provide binary success rewards. The policy is updated using an off-policy reinforcement learning algorithm with a KL-divergence constraint to prevent distribution drift during online updates:

$$\pi^* = \arg\max_{\pi'} \left( \mathbb{E}_{a \sim \pi'}[\hat{A}(s,a)] - \frac{1}{\beta} D_{KL}(\pi' || \pi_\theta) \right), \tag{22}$$

where $\beta$ controls the strength of the constraint between the active policy $\pi'$ and the reference policy $\pi_\theta$. Additionally, the framework incorporates an experience replay buffer with actor confidence filtering to reuse high-fidelity successful trajectories and mitigate catastrophic forgetting of prior knowledge.

## F. Implementation Details

We introduce the details of the implementation of JitRL:

**State Representation.** Efficient retrieval relies on compressing raw observations into representations that capture structural similarity while ignoring irrelevant noise. We design task-specific strategies:

- **WebArena**: In the WebArena environment, raw observations typically consist of full HTML DOM trees and screenshots. While information-rich, these modalities are overly dense and noisy for efficient retrieval. To address this, we design a lightweight representation. First, we use the Regularized URL as a structural proxy. We replace concrete content identifiers (e.g., product names, unique IDs) with generic placeholders. This ensures that different instances of the same page type (e.g., `.../customer/edit/123` and `.../customer/edit/456`) map to a unified "generic view," enabling cross-instance transfer. Second, since URLs are static, we augment them with the **local action history** (e.g., text typed into a search bar) to capture client-side state changes that occur before page reloads.
- **Jericho**: We compress text observations into structured summaries: `Step t: [State: nouns...]` `[Action: verbs...]`. This retains key entities while discarding stylistic flavor text. We further maintain a hierarchical context containing: (1) a global `[SUMMARY]` of objectives, (2) a `[PROGRESS]` list of achieved milestones, and (3) a pruned `[LOCATION]` trajectory that removes unproductive loops.

**Augmented Candidate Set.** To prevent the model from overlooking historically effective actions, we construct an augmented candidate set $\mathcal{C}$. This set merges the LLM's top-$k$ predicted actions ($\mathcal{C}_{LLM}$) with the unique actions found in the retrieved neighborhood $\mathcal{N}(s)$:

$$\mathcal{C} \leftarrow \mathcal{C}_{LLM} \cup \{a_i : (s_i, a_i, G_i) \in \mathcal{N}(s)\}. \tag{23}$$

For any retrieved action $a \notin \mathcal{C}_{LLM}$, we initialize its base logit to a neutral value $z(s,a) = 0$.

**Retrieval Similarity Metric.** We use text-based state representations. To quantify the similarity between the current state $s$ and a memory state $s_i$, we tokenize both representations and compute the **Jaccard Similarity**:

$$J(s, s_i) = \frac{|T(s) \cap T(s_i)|}{|T(s) \cup T(s_i)|}, \tag{24}$$

where $T(\cdot)$ denotes the set of tokens derived from a state.

**Memory Retrieval.** Based on the compressed representations, we employ retrieval strategies designed to balance semantic relevance and structural exactness:

- **WebArena**: We use a two-stage hierarchical matching. We first filter the memory bank by the Regularized URL to locate entries from the same page type. Within this subset, we compute the similarity of the effective state (defined by current filters or input fields) using Jaccard similarity on tokenized representations. This ensures that retrieved experiences share both the same site structure and interactive context.
- **Jericho**: We employ hybrid semantic retrieval using dual vector indexes: a *state index* (current description) and a *history index* (trajectory context). The final retrieval score is a weighted sum (0.75 state + 0.25 history), further refined by Jaccard similarity filtering to ensure sufficient lexical overlap with the query.

**Action Handling.** To ensure that actions retrieved from history are applicable to the current state, we perform normalization and valid-set constraint:

- **WebArena**: Raw actions often contain ephemeral element IDs (e.g., `click('1240')`) that change across sessions. We apply semantic normalization by mapping these IDs to their accessibility tree descriptions (e.g., `click(<combobox[Sort by:]>)`). This converts instance-specific commands into functional semantics recognizable across episodes.
- **Jericho**: To prevent "invalid command" errors common in text games, we explicitly constrain the LLM's output space to the set of valid actions provided by the game engine at each step, ensuring executable policy outputs.

**Advantage Normalization.** To ensure numerical stability across different scales, we normalize the advantage:

$$\tilde{A}(s,a) = \frac{A(s,a)}{\max_{a' \in \mathcal{C}} |A(s,a')| + \epsilon}. \tag{25}$$

## G. Prompt Templates

We provide the complete prompt templates used in JitRL. These prompts are essential for (1) generating candidate actions, (2) evaluating step-wise rewards, and (3) extracting effective trajectory context for memory retrieval. We organize the prompts by task domain.

### G.1. WebArena Prompts

#### G.1.1. ACTION GENERATION PROMPT

The following prompt is used to generate candidate actions from the LLM. The agent receives the current web page state and browsing history, then proposes multiple action options with reasoning.

---

**Action Generation System Prompt**

You are an intelligent web agent that interacts with real web pages on behalf of the user. Your goal is to accurately follow the user's natural language instructions by selecting and executing appropriate web actions. Select promising actions based on the web state and memory of past interactions.
**User's instructions:** {`task_goal`}
**Available Actions:** {`action_space_description`}
**Response Format:** You need to think step-by-step, then provide N potential actions you can take as a web agent.
**IMPORTANT:** When analyzing the state and selecting actions, carefully avoid repeating ineffective patterns:
- If an action failed or gave no progress before, don't try it again in the same context
- Send message to the user once the instruction is fulfilled
**CRITICAL:** `send_msg_to_user()` is a TASK-ENDING action – once called, the task IMMEDIATELY TERMINATES.
You must respond with a JSON object containing:
- `option1`, `option2`, ...: Each option should be a single action command
- `option1_reasoning`, `option2_reasoning`, ...: Explain WHY you chose this action
- `reasoning`: Your overall reasoning about the current state
- `best_action`: The number of the option you think is best

---

#### G.1.2. STEP-WISE REWARD EVALUATION PROMPT

After each episode, we use an LLM-based evaluator to assign step-wise rewards. This prompt instructs the evaluator to score each action based on whether it contributed to the task goal.

---

**Step-wise Reward Evaluation System Prompt**

You are evaluating web agent actions for a training database.
**EVALUATION PROCESS:**
1. Analyze: What happened after this action? What was the result?
2. Determine: Was this action USEFUL or HARMFUL (or neither)?
3. Assess: How CERTAIN are you? (certain / somewhat uncertain / very uncertain)
4. Score: Based on usefulness AND certainty
**SCORING GUIDE** ($-3$ **to** $+3$)**:**
*For USEFUL actions:*

---

- **+3**: Clearly useful AND you are certain
- **+2**: Useful but you are somewhat uncertain
- **+1**: Might be useful but you are very uncertain

*For HARMFUL/USELESS actions:*
- **-3**: Clearly harmful/useless AND you are certain
- **-2**: Harmful/useless but you are somewhat uncertain
- **-1**: Might be harmful/useless but you are very uncertain

*For NEUTRAL actions:*
- **0**: Cannot determine OR no real effect OR effect immediately undone

**MANDATORY FORMAT:**
```
Result: [what happened].
Usefulness: [useful/harmful/neutral].
Certainty: [certain/somewhat
uncertain/very uncertain].
Score: [score] - [repeat/avoid]
```

### G.1.3. TRAJECTORY CONTEXT EXTRACTION PROMPT

For memory retrieval, we extract effective trajectory context that identifies the current state. This prompt uses vision capabilities to analyze screenshots and determine which actions actually define the current state.

---

**Trajectory Context Extraction System Prompt**

You are an expert at extracting effective action sequences to identify unique page states.

**YOUR MISSION:** We want to find similar page states from past browsing sessions to reuse successful actions. We already found pages with the SAME URL, but the same URL can have DIFFERENT STATES (e.g., different filter settings, form values, etc.). Your job: Extract the MINIMAL action sequence that DEFINES the current page state. This action sequence will be used to match against historical states.

**WHAT TO KEEP:**
- Actions whose effects are VISIBLE in the final page
- Actions that DETERMINE what content appears (filters, dropdowns, form inputs)

**WHAT TO REMOVE:**
- Actions whose effects are NOT visible (overwritten/replaced)
- Actions that don't affect final state

**EXAMPLES:**
*Example 1 – Filter defines state:*
Actions: click(Refresh) → select_option(Show By, Year) → click(Refresh)
Final page: Shows "Show By: Year" in dropdown
→ Output: select_option(<combobox[Show By:]>, "Year")
*Example 2 – Overwritten input:*
Actions: fill(From, "2020") → fill(From, "2023")
Final page: Shows "From: 2023"
→ Output: fill(<textbox[From:]>, "2023")

**OUTPUT FORMAT:**
Step 1: Reasoning – Examine final page and check each action's visibility
Step 2: Final Answer – [Action sequence separated by "→", or "No action"]

---

### G.2. Jericho Prompts

### G.2.1. ACTION GENERATION PROMPT

The following prompt is used to generate candidate actions in text-based games. The agent receives the game history, current state, and a list of valid actions from the game engine.

---

**Jericho Action Generation System Prompt**

You are an expert player aiming to complete a text-based adventure game. Points are given for making progress in the game. Select promising actions based on the game state and memory of past interactions.

**EXPLORATION PRIORITY**: When you arrive at a NEW location you haven't fully explored before, you MUST thoroughly explore it FIRST before leaving.

---

**CRITICAL CONSTRAINT**: When REFERENCE ACTIONS are provided, you MUST ONLY choose actions from that list. Any action not in the REFERENCE ACTIONS list is INVALID and will fail. Do NOT create custom actions.

---

**Jericho Action Generation User Prompt**

**GAME HISTORY:**
{summary}
{memory_text}
**RECENT STEPS:**
{recent_history}
**CURRENT STATE:** {current_state}
**TASK:**
1. Analyze your progress: What have you achieved? What's your next objective?
2. **MANDATORY**: Check the REFERENCE ACTIONS list below – you MUST ONLY select from this list
3. Propose $N$ different actions with reasoning
4. For EACH action, provide your confidence as an integer from 0 to 100
**RESPONSE FORMAT (JSON):**

```
{
  "progress_analysis": "What you've achieved and current challenges",
  "next_objective": "Your next goal",
  "reasoning1": "Why this action makes sense",
  "option1": "action command",
  "confidence1": 50,
  ...
  "best_action": 1
}
```

**IMPORTANT:**
• ALL actions MUST be selected EXACTLY from the REFERENCE ACTIONS list
• The sum of all confidence values MUST equal 100
• Pay attention to game hints and clues in state descriptions
• Don't repeat failed actions or create loops (e.g., north→south→north)
**REFERENCE ACTIONS:** {valid_actions}

---

G.2.2. STEP-WISE REWARD EVALUATION PROMPT

After each episode, we evaluate step-wise rewards based on the game's score changes and action effectiveness.

---

**Jericho Step-wise Reward Evaluation Prompt**

You are scoring game actions to build training data for future gameplay.
**PURPOSE:** Rate each action based on its overall impact – positive scores for actions worth repeating, negative for actions to avoid.
**SCORING RULES:**
• **Positive**: Action led to progress or useful discoveries
• **Negative**: Action wasted time, caused loops, or had no benefit
• **Magnitude**: Match the game's typical reward scale (calibrate based on rewards shown in trajectory)
• **Evaluation**: Judge by full consequence chain, not just immediate result
**ANALYSIS:** For each action, explain what happened and why future players should repeat or avoid it.
**JSON FORMAT:**

```
{
  "step_analysis": [
    {
      "step": 0,
      "action": "exact action taken",
      "detailed_reasoning": "What happened after this action
                            and its consequences? (80+ words)",
      "score": 5
    }
  ],
  "overall_assessment": "Key lessons: what worked and what didn't"
}
```

---

### G.2.3. STATE SUMMARIZATION PROMPT

To compress game states for efficient retrieval, we extract key information from each step into a structured format.

---

**Jericho State Summarization Prompt**

You are an expert at summarizing game trajectories. Extract and format key information from each step in a structured way.
**TASK:** Analyze the game trajectory and provide a structured summary.
**FORMAT REQUIREMENTS:**
1. For each step, extract ALL key nouns/objects from the current state IN THE ORDER they appear
2. Extract 1–2 action keywords
3. Format as: `Step X: [State: noun1, noun2, ...] [Action: keyword1, keyword2]`
4. For the LAST step, only output `[State: ...]`, no `[Action: ...]`
**EXAMPLE FORMAT:**
```
Step 0: [State: dark room, stone walls, locked door, rusty key,
        torch, shadows] [Action: take torch]
Step 1: [State: illuminated room, wooden chest, golden lock,
        stone floor, treasure map] [Action: open chest]
Step 2: [State: treasure room, gold coins, silver jewelry,
        exit door, victory banner]
```

---

### G.2.4. HISTORY SUMMARY PROMPT

For cross-episode learning, we generate hierarchical summaries that capture game progress, milestones, and location trajectories. This overall summary will help the generator to understand the current situation, and make better decisions.

---

**Jericho History Summary Prompt**

You are an expert at analyzing game trajectories and creating highly distinctive summaries.
**OUTPUT FORMAT:**
`[SUMMARY]`: Natural language summary describing:
• The game's objective or goal (inferred from trajectory)
• Current progress toward that goal
• Key accomplishments so far
• Current situation/status
`[PROGRESS]`: List milestones in format ✓ `M#: <action>→<key object/result>`
• ONLY record steps where the score increased
• If NO steps have score increases, output "No Progress"
`[LOCATION]`: Track location changes in format `Location: A→B→C`
• Extract location names from state descriptions
• Only record when location actually changes
• CRITICAL: IGNORE unproductive loops – if the player goes back and forth WITHOUT score increases, skip those movements
**EXAMPLES:**
*Good Summary:* "The game's objective is to escape the haunted mansion. So far, the player has found a key in the library and unlocked the basement door, earning 15 points. Currently in the dark basement, the player needs to find a light source."
*Good Progress:* ✓ `M1: enter→library`
*Good Location:* `Location: entrance→library→basement`
(Not: `room A→room B→room A→room B` – this is an unproductive loop)

---

## H. Hyperparameters

We provide detailed hyperparameter settings for the Jericho experiments in Table 10, and detailed hyperparameter settings for the WebArena experiments in Table 11.

## I. Comparison of WebRL and JitRL under Identical Settings

To further isolate the impact of the proposed JitRL, we conducted a controlled experiment comparing JitRL with WebRL using the identical base model, **Llama 3.1 8B Instruct**, and evaluated both on-the-fly on the WebArena-Lite subset. To ensure a strictly fair comparison, we maintained identical experimental settings for both methods: for each task, the agent is granted five attempts. The trajectories from previous attempts are utilized by both models to improve subsequent

*Table 10.* Hyperparameters for Jericho experiments.

| Category | Parameter | Value |
|---|---|---|
| *Environment Settings* | | |
| | Step limit per episode | 60 |
| | Number of episodes | 50 |
| *LLM Configuration* | | |
| | Temperature | 0.8 |
| | Candidate actions ($|\mathcal{C}|$) | 3 |
| *Value Estimation* | | |
| | Discount factor ($\gamma$) | 0.5 |
| | Retrieval neighbors ($k$) | 10 |
| | Similarity threshold ($r$) | 0.95 |
| *Exploration* | | |
| | Exploration rate ($\lambda$) | 0.65 |
| | UCB bonus ($\alpha$) | 5 |

performance: JitRL stores these trajectories in its *dynamic memory* for inference-time retrieval, whereas WebRL uses them for *online policy updates* via reinforcement learning.

As summarized in Table 12, JitRL outperforms WebRL in the majority of domains and achieves a higher overall average success rate. While both methods attempt to learn from experience during the test phase, the performance gap highlights the inherent limitations of weight-update-based RL in low-data, on-the-fly scenarios. We attribute WebRL's inferior performance to the fact that standard RL algorithms typically require a vast amount of environment interactions and high-quality samples to achieve stable convergence. In contrast, JitRL's inference-time optimization bypasses the instability of gradient-based updates in small-sample regimes, allowing for more immediate and robust adaptation to complex web environments.

# J. Additional Controlled Comparisons

## J.1. Controlled Comparison on WebArena with the Same Backbone

We conduct a controlled experiment using the same Llama-3.1-70B-Instruct backbone for all methods, following the same protocol as WebRL: both JitRL and WebRL use identical training tasks (WebArena tasks excluding WebArena-Lite) and are evaluated on the held-out WebArena-Lite (165 tasks) with no test-set adaptation. During training, JitRL collects experience memory without gradient updates; during evaluation, it retrieves from this pre-collected memory. As shown in Table 13, JitRL substantially outperforms SFT and approaches WebRL's performance at a fraction of the cost ($200 vs. $9,900). We note that this setting — with a large offline training corpus and no test-time adaptation — is specifically favorable to weight-update methods; JitRL is primarily designed for the continual learning regime where such offline datasets are unavailable.

## J.2. Same-Backbone Comparison on Jericho

We compare JitRL and GRPO using the same Qwen3-32B model on all three Jericho games. Both methods operate in the continual learning setting: GRPO performs online gradient updates after each episode, while JitRL accumulates experience memory. As shown in Table 14, JitRL outperforms GRPO across all three games, with the most pronounced difference observed on Zork1.

# K. Case Studies on Jericho

Table 17 presents analogous correction patterns in text-based games. We identify three key mechanisms. First, regarding **Exploration Override** (Library), the base model defaults to generic investigative actions ("examine desk"), but memory encodes that direct task-relevant interactions ("give ID to attendant") yield immediate rewards. Second, for **Game Mechanics**

*Table 11.* Hyperparameters for WebArena experiments.

| Category | Parameter | Value |
|---|---|---|
| *Environment Settings* | | |
| | Step limit per episode | 10 |
| | Number of episodes | 50 |
| | Random seed | 0 |
| *LLM Configuration* | | |
| | Temperature | 0.8 |
| | Candidate actions ($|\mathcal{C}|$) | 3 |
| *Value Estimation* | | |
| | Discount factor ($\gamma$) | 0.1 |
| | Retrieval neighbors ($k$) | 10 |
| | Similarity threshold ($r$) | 0.8 |
| *Memory Retrieval* | | |
| | Task similarity threshold | 0.27 |
| | History weight | 0.7 |
| | Task weight | 0.3 |
| *Exploration* | | |
| | Exploration rate ($\lambda$) | 0.05 |
| | UCB bonus ($\alpha$) | 5 |

*Table 12.* Final success rate comparison (%) between WebRL and JitRL on WebArena-Lite using the same base model and training data.

| Method | Admin | GitLab | Map | Reddit | Shopping | Avg |
|---|---|---|---|---|---|---|
| WebRL | **38.89** | 23.53 | 9.68 | 50.00 | 17.39 | 27.27 |
| **JitRL** | 28.57 | **30.00** | **20.00** | **53.33** | **36.96** | **32.97** |

(Zork1), JitRL learns non-obvious puzzle solutions—in the Loud Room, the intuitive "take platinum bar" fails due to deafening noise, while the counterintuitive "echo" command quiets the room and enables collection. Third, concerning **Safety-Critical Sequencing** (Zork3), the base model attempts a direct "climb down" at the cliff, but memory shows that "tie rope to railing" must precede descent to avoid fatal falls.

## L. Impact of State Representation

We compare our structured text state summary (**Text**) against state embedding (**Embedding**) on Library and Zork1. Figure 5 shows that the Text achieves superior stability and performance, whereas Embedding exhibits significant fluctuation. We attribute this to the low discriminative power of dense embeddings in text games; they often assign high similarity to topologically distinct but descriptively similar states. This ambiguity leads to the retrieval of mismatched memories, introducing noise that degrades policy improvement.

## M. Step Analysis

Table 18 presents the average number of steps required by the agent to complete tasks across different website domains. The results reveal notable differences in task complexity among the domains. The Map website poses the greatest challenge, requiring an average of 9.39 steps per task, as map-based navigation typically involves sequential operations such as searching locations, adjusting views, and extracting information. In contrast, Reddit and Admin websites present relatively simpler navigation patterns, averaging only 4.57 and 5.55 steps, respectively, as these platforms often allow more direct access to target content.

*Table 13.* Controlled comparison on WebArena-Lite (Final success rate %) using the same `Llama-3.1-70B-Instruct` backbone, with separate training and evaluation phases.

| Method | Admin | GitLab | Map | Reddit | Shopping | Average | Cost |
|---|---|---|---|---|---|---|---|
| Llama-3.1-70B | 10.50 | 16.70 | 17.10 | 20.00 | 4.40 | 12.70 | – |
| SFT | 20.00 | 20.00 | 26.70 | 52.60 | 13.30 | 23.00 | $640 |
| **JitRL** | 47.22 | 38.24 | 25.81 | 58.33 | **34.78** | 40.88 | $200 |
| WebRL | **58.33** | **47.06** | **32.26** | **62.50** | 30.43 | **46.06** | $9,900 |

*Table 14.* Same-backbone comparison on Jericho (Qwen3-32B, both methods adapt on the test set, Avg score over 50 episodes).

| Method | Library | Zork1 | Zork3 |
|---|---|---|---|
| GRPO | 13.6 | 16.2 | 1.1 |
| **JitRL** | **20.8** | **42.1** | **2.3** |

# N. Ablation on Exploration Rate $\lambda$

We ablate the exploration rate $\lambda$ on both Jericho (three games: Library, Zork1, Zork3; averaged over 50 episodes) and WebArena (five website domains; averaged over 5 episodes per task). We vary $\lambda \in \{0, 0.25, 0.50, 0.65, 0.75\}$ for Jericho and $\lambda \in \{0, 0.05, 0.25, 0.50, 0.75\}$ for WebArena, keeping all other hyperparameters fixed. As shown in Tables 19 and 20, performance is stable across a reasonable range of both parameters. WebArena favors a smaller $\lambda$ since its structured action space requires less exploration, while Jericho benefits from a larger $\lambda$ due to its combinatorially large action space, where extensive exploration is essential to discover effective action sequences.

# O. Sensitivity to UCB Bonus $\alpha$

We ablate the UCB bonus $\alpha$ on both Jericho (three games: Library, Zork1, Zork3; averaged over 50 episodes) and WebArena (five website domains; averaged over 5 episodes per task), varying $\alpha \in \{3, 5, 7, 10\}$ while keeping all other hyperparameters fixed. As shown in Tables 21 and 22, performance is stable across a reasonable range of $\alpha$, with $\alpha = 5$ achieving the best overall balance between exploration incentive and over-optimism.

# P. Scalability of JitRL

We evaluate JitRL's scalability on the Library game from Jericho over 50 episodes, measuring how retrieval latency and average game score evolve as the memory bank grows. We partition the memory into bins of 500 entries and report the

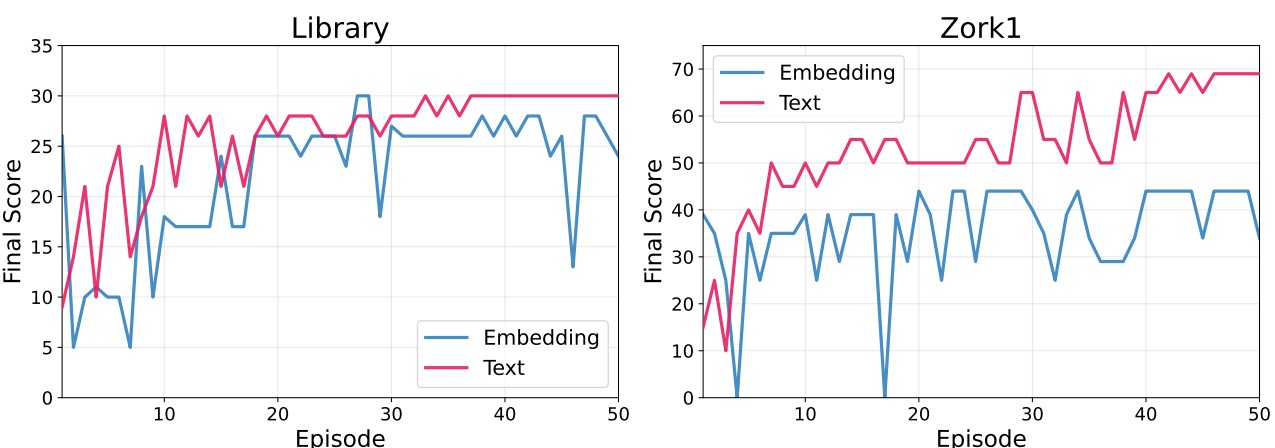

*Figure 5.* Performance Comparison of Text-based and Embedding-based State Representations.

*Table 15.* Unified pipeline comparison on Jericho (Avg score over 50 episodes).

| Method | Library | Zork1 | Zork3 |
|---|---|---|---|
| EvoTest (best baseline) | 21.5 | 46.8 | 2.6 |
| JitRL (unified) | 24.7 | 51.8 | 3.3 |
| **JitRL** | **25.9** | **53.0** | **3.1** |

*Table 16.* Unified pipeline comparison on WebArena (Avg success rate %).

| Method | Admin | GitLab | Map | Reddit | Shopping | Average |
|---|---|---|---|---|---|---|
| EvoTest (best baseline) | 44.51 | 37.94 | 33.59 | 51.78 | 26.67 | 39.24 |
| JitRL (unified) | 52.17 | 41.18 | 35.23 | 58.14 | 43.33 | 46.01 |
| **JitRL** | **52.31** | **40.78** | **37.66** | **57.64** | **41.67** | **46.98** |

average retrieval time (in milliseconds) and average score within each bin. As shown in Table 23, JitRL's performance continuously improves as memory grows, while retrieval overhead remains negligible compared to LLM inference time.

## Q. GRPO Hyperparameter Sweep and Comparison with JitRL

### Q.1. Clarification on Rollout Count vs. Batch Size

In GRPO, each problem instance is rolled out $G$ times, and the advantage for each trajectory is computed via group-relative normalization over all trajectories sharing the same instance:

$$\hat{A}_i = \frac{r_i - \text{mean}(\{r_j\}_{j \in \text{group}})}{\text{std}(\{r_j\}_{j \in \text{group}})}. \tag{26}$$

Since all training data in our Jericho setting originates from the same initial game observation (analogous to a single prompt), increasing rollout count $G$ and increasing batch size both raise the total number of trajectories per gradient step, but differ in the scope of group normalization:

- **Increasing rollout count** (e.g., $G=64$, batch size=1): All 64 trajectories share one group ID, so group-relative normalization is performed jointly over all 64 trajectories, yielding a smoother advantage estimate.
- **Increasing batch size** (e.g., $G=8$, batch size=8): Each copy of the instance receives an independent group ID, creating 8 independent groups of 8 rollouts. Normalization is performed within each group of 8, making the advantage more sensitive to local variation.

### Q.2. Experimental Setup

We conduct experiments on Zork1 using Qwen3-32B (bf16) trained with GRPO for 50 training steps, matching the number of episodes used in JitRL evaluation. After each training step, we evaluate the model with a single rollout and record the score. We sweep four hyperparameters — rollout count $G$, batch size, PPO epochs, and learning rate — varying one axis at a time while fixing the others. All experiments are run on $4\times$ NVIDIA H200 GPUs with tensor parallelism of 4, a maximum of 80 turns per episode, a maximum response length of 8,192 tokens, and KL coefficient 0.001 (low-variance KL).

### Q.3. Results

Table 24 summarizes the validation mean, max, and min scores across all swept configurations, along with the total number of training episodes consumed by each setting.

### Q.4. Analysis

As shown in Table 24, the best GRPO configuration (rollout=8, batch size=8, PPO epochs=1, lr=1e-5) achieves a mean validation score of 40.7 and a maximum of 55 on Zork1, with a clear upward learning curve that stabilizes near 55 in the

*Table 17.* **Qualitative Analysis of Policy Improvement on Jericho.** We select one representative case from each text-based game showing how JitRL modifies decision-making. **Base** and **JitRL** denote the logits before and after the memory-based update. The results highlight JitRL's ability to override exploratory defaults, learn game-specific mechanics, and prioritize high-reward actions.

| State / Context | Candidate Action | Base | JitRL | Mechanism Explanation |
|---|---|---|---|---|
| Library: Lobby (Attendant present) | examine desk
give ID to attendant | **0.85**
0.45 | 0.35
**1.65** | The base model defaults to exploratory actions.
Memory encodes that this action immediately yields +5 points and unlocks the rare books room. |
| Zork1: Loud Room (Deafening noise) | take platinum bar
echo | **0.92**
0.30 | 0.42
**1.50** | Greedy treasure-grabbing seems intuitive, but fails
in the noisy room. JitRL learns that "echo" quiets the room, enabling treasure collection. |
| Zork3: Cliff (Holding rope) | climb down
tie rope to railing | **0.90**
0.40 | 0.30
**1.60** | Direct descent without preparation leads to death.
Memory encodes that securing the rope first enables safe descent and progression. |

*Table 18.* Average Steps per Website.

| Website | GitLab | Map | Reddit | Shopping | Admin | Overall |
|---|---|---|---|---|---|---|
| **Avg Steps** | 5.83 | 9.39 | 4.57 | 5.55 | 6.20 | **6.28** |

final 20 steps. This is competitive with JitRL's mean score of 53.0 using the same Qwen3-32B backbone.

However, this result comes at a substantial sample cost. Over 50 steps, JitRL requires exactly **50 trajectories** in total. In contrast, GRPO consumes $50 \times G \times$ batch size trajectories per run. The best GRPO configuration alone requires **3,200 training episodes** — over **64×** **more** than JitRL. Even the original GRPO setting reported in the main paper (rollout=8, batch size=1) requires 400 episodes, **8×** **more** than JitRL. While GRPO with extensive hyperparameter tuning can approach JitRL's performance, this comes at a substantially higher sample cost, further highlighting JitRL's efficiency advantage.

## R. Cost Analysis.

The training of **WEBRL**, based on the `Llama-3.1-70B-Instruct` backbone, requires approximately 154 hours of computation on a 16 × NVIDIA H200 GPU cluster (distributed across 2 nodes). This intensive online reinforcement learning process is structured into two primary stages: the initial Supervised Fine-Tuning (SFT) phase, which accounts for 10 hours to establish foundational web-navigation capabilities, followed by an iterative 8-phase Reinforcement Learning cycle. Within each RL phase, the Self-Evolving Task Generation and Filtering requires 8 hours (1 hour per phase), while the Online Rollout and Interaction phase for generating trajectories takes approximately 48 hours (6 hours per phase). The Reward Labeling (utilizing an 8B ORM) and the heavy Actor-Critic Optimization (managing 70B parameters with a 16,384 cutoff length) require 2 hours (15 mins per phase) and 86 hours (10.75 hours per phase), respectively. Based on a standard market rate of $64 per hour for 2 nodes, the total estimated expenditure for a single complete 70B training cycle is approximately $9,856. The operational costs for JitRL and other training-free baselines are calculated based on token consumption through the OpenRouter API, which serves as the uniform billing standard for these inference-based methods.

*Table 19.* Ablation of $\lambda$ on Jericho (Avg score over 50 episodes).

| $\lambda$ | Library | Zork1 | Zork3 |
|------|---------|-------|-------|
| 0    | 20.8    | 42.1  | 2.1   |
| 0.25 | 24.1    | 48.6  | 2.6   |
| 0.50 | 24.9    | **52.7** | **3.1** |
| **0.65** | **25.9** | 53.0 | **3.1** |
| 0.75 | 25.7    | 51.4  | 2.9   |

*Table 20.* Ablation of $\lambda$ on WebArena (Avg success rate %, full).

| $\lambda$ | Admin | GitLab | Map | Reddit | Shopping | Average |
|------|-------|--------|-----|--------|----------|---------|
| 0    | 50.77 | 39.51 | 36.41 | 55.81 | 40.42 | 45.38 |
| **0.05** | 52.31 | 40.78 | **37.66** | 57.64 | **41.67** | **46.98** |
| 0.25 | **52.75** | **41.18** | 37.19 | **57.89** | 41.25 | 46.52 |
| 0.50 | 49.01 | 38.04 | 34.38 | 53.49 | 37.08 | 43.36 |
| 0.75 | 46.15 | 36.27 | 32.81 | 51.94 | 35.42 | 41.32 |

*Table 21.* Sensitivity of $\alpha$ on Jericho (Avg score over 50 episodes).

| $\alpha$ | Library | Zork1 | Zork3 |
|------|---------|-------|-------|
| 3    | 23.8    | 50.5  | 2.7   |
| **5** | **25.9** | 53.0 | **3.1** |
| 7    | 24.6    | **54.2** | 2.8 |
| 10   | 23.4    | 49.2  | 2.6   |

*Table 22.* Sensitivity of $\alpha$ on WebArena (Avg success rate %).

| $\alpha$ | Admin | GitLab | Map | Reddit | Shopping |
|------|-------|--------|-----|--------|----------|
| 3    | **53.48** | 39.80 | 35.78 | **58.45** | 40.52 |
| **5** | 52.31 | **40.78** | **37.66** | 57.64 | **41.67** |
| 7    | 49.02 | 39.22 | 34.50 | 55.04 | 38.54 |
| 10   | 46.52 | 37.45 | 33.03 | 53.02 | 36.98 |

*Table 23.* Inference overhead and performance as memory grows (Library, 50 episodes).

| Memory Size (# entries) | Retrieval (ms) | Avg Score |
|-------------------------|----------------|-----------|
| 0–500     | 15–22 | 18.1 |
| 500–1000  | 26–27 | 25.1 |
| 1000–1500 | 29    | 27.2 |
| 1500–2000 | 38    | 29.2 |
| 2000–2500 | 47    | 30.0 |

---

**Algorithm 1** JitRL: Test-Time Policy Optimization

---

**Input:** LLM $\pi_\theta$, hyperparameters $k, \beta, \lambda, \alpha, \gamma$
**Initialize:** Memory $\mathcal{M} \leftarrow \emptyset$
**for** each episode **do**
   Initialize trajectory $\tau \leftarrow []$
   **while** episode not done **do**
      Observe state $o$; abstract to $s \leftarrow \text{AbstractState}(o)$
      *// Step 1: Retrieve similar experiences*
      $\mathcal{N}(s) \leftarrow \text{Top-}k$ neighbors from $\mathcal{M}$ by $\text{Jaccard}(s, s_i)$
      *// Step 2: Estimate state value (baseline)*
      $V(s) \leftarrow \frac{1}{|\mathcal{N}(s)|} \sum_{(s_i, a_i, G_i) \in \mathcal{N}(s)} G_i$
      *// Step 3: Construct augmented candidate set*
      Get LLM candidates $\mathcal{C}_{\text{LLM}}$ with logits $z(s, a)$
      $\mathcal{C} \leftarrow \mathcal{C}_{LLM} \cup \{a_i : (s_i, a_i, G_i) \in \mathcal{N}(s)\}$.
      **for** $a \in \mathcal{C} \setminus \mathcal{C}_{\text{LLM}}$ **do**
         $z(s, a) \leftarrow 0$                       *// Initialize logit for memory-only actions*
      **end for**
      *// Step 4: Estimate value for each candidate*
      **for** $a \in \mathcal{C}$ **do**
         $\mathcal{N}(s, a) \leftarrow \{(s_i, a_i, G_i) \in \mathcal{N}(s) : a_i = a\}$
         **if** $|\mathcal{N}(s, a)| > 0$ **then**
            $Q(s, a) \leftarrow \frac{1}{|\mathcal{N}(s, a)|} \sum_{(s_i, a_i, G_i) \in \mathcal{N}(s, a)} G_i$       *// Known action*
         **else**
            **if** $\text{rand}() < \lambda$ **then**
               $Q(s, a) \leftarrow V(s) + \alpha/|\mathcal{N}(s)|$       *// Exploration bonus*
            **else**
               $Q(s, a) \leftarrow 0$                    *// Neutral value*
            **end if**
         **end if**
      **end for**
      *// Step 5: Compute normalized advantage*
      **for** $a \in \mathcal{C}$ **do**
         $A(s, a) \leftarrow Q(s, a) - V(s)$
      **end for**
      $A_{\max} \leftarrow \max_{a' \in \mathcal{C}} |A(s, a')| + \epsilon$
      **for** $a \in \mathcal{C}$ **do**
         $\tilde{A}(s, a) \leftarrow A(s, a)/(A_{\max} + \epsilon)$
      **end for**
      *// Step 6: Closed-form policy update*
      **for** $a \in \mathcal{C}$ **do**
         $z'(s, a) \leftarrow z(s, a) + \beta \cdot \tilde{A}(s, a)$
      **end for**
      Sample $a \sim \text{Softmax}(z')$; append $(s, a)$ to $\tau$
   **end while**
   *// Step 7: Memory update after episode ends*
   $\{r_t\}_{t=0}^{T} \leftarrow \text{Evaluator}(\tau)$
   **for** $t = 0$ **to** $T$ **do**
      $G_t \leftarrow \sum_{u=t}^{T} \gamma^{u-t} r_u$
      $\mathcal{M} \leftarrow \mathcal{M} \cup \{(s_t, a_t, G_t)\}$
   **end for**
**end for**

---

*Table 24.* GRPO hyperparameter sweep on Zork1 (Qwen3-32B, 50 training steps).

| Rollout | Batch Size | PPO Epochs | LR | Val Mean | Val Max | Val Min |
|---------|-----------|-----------|------|----------|---------|---------|
| 8 | 1 | 1 | 1e-6 | 16.2 | 35 | 0 |
| 16 | 1 | 1 | 1e-6 | 13.6 | 40 | 0 |
| 32 | 1 | 1 | 1e-6 | 12.0 | 40 | -5 |
| 8 | 1 | 2 | 1e-6 | 10.4 | 40 | 0 |
| 8 | 1 | 4 | 1e-6 | 17.1 | 40 | 5 |
| 8 | 1 | 4 | 1e-5 | 25.5 | 45 | -5 |
| **8** | **8** | **1** | **1e-5** | **40.7** | **55** | 5 |

