# OpenReview forum: "Just-In-Time Reinforcement Learning: Continual Learning in LLM Agents Without Gradient Updates"
_ICML.cc/2026/Conference — ICML 2026 spotlight_

### Official Review · Reviewer_ZEPB · 2026-03-03

**Soundness:** 2
**Presentation:** 3
**Significance:** 3
**Originality:** 4
**Overall Recommendation:** 4
**Confidence:** 4

**Summary:**

This paper proposes a parameter-free approach for constructing evolving LLM-based agent systems through the use of experience memory. Inspired by reinforcement learning, the method estimates advantages from past experiences on the fly and adjusts the action logits produced by the LLM accordingly. The framework enables adaptive behavior without modifying model parameters. The paper also provides theoretical analysis of the proposed method and presents experimental results to evaluate its effectiveness.

**Compliance With Llm Reviewing Policy:**

Affirmed.

**Final Justification:**

The additional controlled comparisons, same-backbone experiments, and clearer discussion of limitations adequately address my main concerns and strengthen the empirical support of the paper

Overall, I find the work sound and better validated after the rebuttal, and I update my score from 3 to 4 accordingly.

**Key Questions For Authors:**

1. In the main experiments, JitRL uses a different LLM backbone from the weight-update baselines. Can results be provided with a unified backbone to ensure a fair comparison?
2. The multi-run evaluation setting appears not directly comparable to the weight-update baselines, as it effectively allows JitRL to adapt on the test set while the baselines do not. How do the authors justify this setup, and can a strictly controlled comparison be provided?

The paper includes a more controlled comparison with WebRL in Appendix I. Moving this result to the main paper and conducting similarly fair comparisons for the Jericho experiments would strengthen the empirical quality. Clarification on these points will significantly impact my overall evaluation of the paper.

**Limitations:**

No. The paper does not sufficiently discuss the methodological limitations. A clearer analysis of the approach’s assumptions, potential failure modes, and practical constraints would strengthen the work. The potential negative societal impacts are properly discussed in this paper.

**Strengths And Weaknesses:**

### Strengths

1. The method is theoretically grounded and supported by comprehensive experimental evaluation, demonstrating its effectiveness.
2. The paper is clearly written and well organized, making the technical contributions easy to follow.
3. The work addresses on-the-fly adaptation in LLM agent systems and introduces a novel approach that leverages non-parametric experience to adjust logits, offering a meaningful perspective for adaptive LLM systems.

### Weaknesses

1. Several experimental settings raise concerns about fairness. In the main paper, JitRL uses a different LLM backbone from the weight-update baselines (SFT, WebRL, GRPO), which weakens the validity of the comparisons.
2. The multi-run evaluation setting is not directly comparable to the baselines and effectively allows adaptation on the test set, leading to potential unfairness.
3. A more controlled comparison appears in Appendix I but is not included in the main paper. Similar fair comparisons are missing for Jericho experiments.
4. The paper lacks comparison with prior RL-inspired, parameter-free LLM memory approaches, such as Rememberer [1] and ICPI [2], as well as more recent work in this line of research.

[1] *Large Language Models Are Semi-Parametric Reinforcement Learning Agents*. NeurIPS 2023.

[2] *Large Language Models Can Implement Policy Iteration*. NeurIPS 2023.

---

> ### Author Rebuttal · Authors · 2026-03-29
>
> Thank you for your constructive feedback! We address your concerns below.
>
> > ### W1–W3 & Q1–Q3: Concern of backbone difference, multi-run fairness, and missing controlled comparisons for Jericho.
>
> Our WebRL and GRPO baselines **follow the training methods and backbones proposed in prior works** [1,2], while JitRL uses Gemini-2.5-flash because training-free methods can leverage powerful closed-source API models. This is itself an advantage over weight-update methods, which are restricted to open-weight models (typically smaller in scale).
>
> We further provide controlled comparisons using the **same backbone** where **both methods adapt on the test set**.
>
> - For **WebArena**, Appendix I (Table 12) has already provided such a comparison. Under these conditions, **JitRL achieves 32.97% vs. WebRL's 27.27%** success rate.
>
> - For **Jericho games**, the GRPO results in Table 3 already use test-set adaptation — GRPO performs online gradient updates after each episode on the test games. To further address the backbone concern, we provide a same-backbone (Qwen3-32B) comparison. As shown in the table, **JitRL outperforms GRPO on all three games**.
>
>
> **Table: Same-backbone comparison on Jericho (Qwen3-32B, both methods adapt on the test set, Avg score over 50 episodes)**
>
> |Method| Library | Zork1| Zork3 |
> |-| -| -| -|
> |GRPO|13.6|16.2|1.1|
> |**JitRL**|**20.8**| **42.1**| **2.3** |
>
> We also conducted a comparison under a **larger budget, non-test-adaptation** setting. We use the same Llama-3.1-70B backbone, where both methods train/collect experience on a separate training set, then evaluate on the held-out WebArena-Lite without any test-set adaptation. This setting provides large-scale training data and no test-set adaptation, which is favorable for weight-update methods. As shown in the table below, even in this setting, **JitRL achieves competitive performance at only ~2% of WebRL's training cost** (\$200 vs. \\$9,900 USD) — though **JitRL is specifically designed for the continual learning setting, not this offline regime**.
>
> **Table: Controlled comparison on WebArena-Lite with same backbone (Llama-3.1-70B) and non-test-adaptation (Final success rate %)**
>
> |Method|Admin|GitLab|Map|Reddit|Shopping|Average|Cost|
> |-|-|-|-|-|-|-|-|
> |SFT|20.00|20.00|26.70|52.60|13.30|23.00|~\$640|
> |JitRL|47.22|38.24|25.81|58.33|**34.78**|40.88|**~\$200**|
> |WebRL|**58.33**|**47.06**|**32.26**|**62.50**|30.43|**46.06**|~\$9,900|
>
> We will include all these results into our main paper.
>
> > ### Limitations: Concern of discussion of the limitations.
>
> We identify two key limitations:
>
> 1. **Dependence on the frozen base model.** JitRL can only re-weight actions the base model already proposes; it cannot discover entirely new actions. Additionally, JitRL relies on the LLM evaluator to produce reasonable step-wise rewards — if the evaluator misattributes credit, the advantage estimates will be inaccurate.
> 2. **Less suited for tasks where key information is hard to represent in text.** JitRL's state representation and retrieval operate on text, so for tasks where the critical patterns are difficult to express textually — e.g., spatial reasoning (chess board positions) or time-series forecasting — text-based retrieval may fail to capture the relevant similarity between states.
>
> We will add these limitations in the revision.
>
> > ### Weakness 4: Comparison with Rememberer and ICPI.
>
> The key distinction among JitRL, ICPI, and Rememberer are:
>
> - **Rememberer** appends retrieved experiences to the prompt as action advice; the LLM interprets this via in-context learning.
> - **ICPI** requires the LLM to predict future transitions for each candidate action to estimate Q-values. In complex environments, accurately predicting state transitions from a few in-context examples is extremely difficult, and prediction errors accumulate over steps, making the Q-value estimates unreliable.
> - **JitRL** uses an LLM evaluator to review the complete trajectory after each episode, providing more accurate reward signals over the full horizon. These real rewards are used to directly compute advantages and modulate logits, with theoretical grounding as the closed-form solution to KL-constrained policy optimization.
>
> We compare ICPI and REmemberer with JitRL using Gemini-2.5-flash. As shown in the table below, JitRL outperforms both baselines across all games.
>
> **Table: Comparison with Rememberer and ICPI on Jericho (Avg score %)**
>
> | Method     | Library | Zork1 | Zork3 |
> | ---------- | ------- | ----- | ----- |
> | ICPI       | 23.8   | 28.1 | 2.2  |
> | Rememberer | 23.5  | 48.0 | 2.0 |
> | JitRL  | **25.9**| **53.0**| **3.1** |
>
> We sincerely thank the reviewer for the valuable feedback. We will incorporate all the discussions and experimental results above into the revised paper.
>
> [1] WebRL: Training LLM Web Agents via Self-Evolving Online Curriculum Reinforcement Learning
>
> [2] EvoTest: Evolutionary Test-Time Learning for Self-Improving Agentic Systems

---

> > ### Author Rebuttal · Reviewer_ZEPB · 2026-04-03
> >
> > Thank the authors for their detailed rebuttal. Based on their response, I will update my recommendation accordingly.

---

> > > ### Author Response · Authors · 2026-04-03
> > >
> > > We sincerely thank you for the positive feedback and for raising the score! Your critical insights have been essential to the refinement of our manuscript, and we appreciate your support.

---

### Official Review · Reviewer_enCW · 2026-03-07

**Soundness:** 2
**Presentation:** 3
**Significance:** 2
**Originality:** 3
**Overall Recommendation:** 4
**Confidence:** 3

**Summary:**

This paper proposes Just-In-Time Reinforcement Learning (JitRL), a retrieval-based RL framework that enables continual adaptation without gradient-based parameter updates. Instead of updating model parameters, the method maintains a trajectory memory and retrieves relevant experiences to estimate action advantages at inference time. These advantage estimates are then used to adjust the model’s output logits.

**Compliance With Llm Reviewing Policy:**

Affirmed.

**Final Justification:**

My main concerns have been addressed after the rebuttal. I update my score from 3 to 4 accordingly.

**Key Questions For Authors:**

1. What is the additional inference overhead introduced by the retrieval and advantage estimation steps compared to standard LLM inference?
2. How does the proposed method scale as the trajectory memory bank grows, both in terms of retrieval efficiency and performance?
3. How sensitive is the performance of JitRL to the choice of the exploration bonus parameter α?

**Limitations:**

yes

**Strengths And Weaknesses:**

Strengths
1. The proposed method enables continual learning in LLM Agents without gradient updates.
2. The experimental results demonstrate improved performance.
3. The paper is well-written and easy to follow.

Weaknesses
1. The method’s performance relies on converting observations into state representations, yet the design is tailored to task-specific strategies, limiting potential generalization.
2. The proposed method introduces additional inference overhead due to trajectory retrieval and advantage estimation. While the paper highlights training-free benefits, the experiments do not quantify or discuss the impact of this overhead on latency and practical deployment.
3. The scalability of the trajectory memory bank is not discussed, particularly how performance and retrieval efficiency are affected as the memory size grows.

---

> ### Author Rebuttal · Authors · 2026-03-29
>
> Thank you for your constructive feedback! We address your concerns below.
>
> > ### W1: Task-specific state representations limit potential generalization.
>
> **The core algorithm is entirely identical** across domains. The state representation for WebArena, URL normalization, is a simple and efficient shortcut we discovered — the normalized URL captures page type and structure without needing an additional LLM summarization. However, it is not the only viable approach.
>
> To directly test this, **we implemented a unified pipeline** where both domains use: (1) the same state representation method (LLM-based summarization), (2) the same retrieval pipeline (pre-filtering + similarity compute), and (3) the same LLM evaluator. The only background provided to the LLM is minimal domain context, which is comparable to the background context that WebRL [2] and EvoTest [3] also provide to their models.
>
> **Jericho (Avg score):**
>
> | Pipeline                | Library  | Zork1    | Zork3   |
> | ----------------------- | -------- | -------- | ------- |
> | JitRL                   | **25.9** | **53.0** | 3.1 |
> | JitRL (unified)         | 24.7     | 51.8     | **3.3** |
> | Best baseline (EvoTest) | 21.5     | 46.8     | 2.6     |
>
> **WebArena (Avg success rate %):**
>
> | Pipeline | Admin | GitLab | Map | Reddit | Shopping | Average |
> | - | - | - | - | - | - | - |
> | JitRL | **52.31** | 40.78 | **37.66** | 57.64 | 41.67 | **46.98** |
> | JitRL (unified) | 52.17 | **41.18** | 35.23 | **58.14** | **43.33** | 46.01 |
> | Best baseline (EvoTest) | 44.51 | 37.94 | 33.59 | 51.78 | 26.67 | 39.24 |
>
> As shown in the tables above, the unified pipeline is still comparable to our previous results, while still **outperforming all baselines**. This confirms that **JitRL's gains primarily stem from the advantage estimation and logit update mechanism**.
>
> > ### W2 & W3 / Q1 & Q2: Inference overhead and memory scalability.
>
> We conducted an ablation on Library (Jericho) across 50 episodes, measuring per-step latency for both **retrieval** and **advantage estimation** as the memory bank grows naturally over episodes.
>
> **Table: Inference overhead and performance as memory grows (Library, 50 episodes)**
>
> | Memory Size (# entries) | Retrieval (ms) | Advantage Est. (ms) | Avg Score |
> | - | - | - | - |
> | 0–500                   | 15–22          | <0.02               | 18.1      |
> | 500–1000                | 26–27          | <0.02               | 25.1      |
> | 1000–1500               | 29             | <0.02               | 27.2      |
> | 1500–2000               | 38             | <0.02               | 29.2      |
> | 2000–2500               | 47             | <0.02               | 30.0 (max) |
>
> **Key findings:**
>
> 1. **Performance continuously improves as memory grows.** Average score increases from 18.1 (0–500 entries) to 30.0 (2000–2500 entries, the maximum score), a **66% relative improvement**.
> 2. **Retrieval and advantage estimation overhead is negligible.** Per-step retrieval latency ranges from 15ms to 47ms, while advantage estimation is pure arithmetic and takes <0.02ms. Compared to a single LLM API call, the total overhead constitutes **<2%**.
> 3. **Overhead scales gracefully with memory size.** Retrieval latency grows linearly from 15ms to 47ms as memory expands from 0 to 2500 entries. For longer deployments, adopting Faiss [1] (which supports billion-scale similarity search in milliseconds) can **further reduce retrieval latency**.
>
>
>
> > ### Q3: How sensitive is performance to the exploration bonus α?
>
> We conducted a sensitivity analysis varying α ∈ {3, 5, 7, 10}.
>
> **Table: Sensitivity of α on Jericho (Avg score)**
>
> | α     | Library  | Zork1    | Zork3   |
> | ----- | -------- | -------- | ------- |
> | 3     | 23.8     | 50.5     | 2.7     |
> | 5 | **25.9** | 53.0     | **3.1** |
> | 7     | 24.6     | **54.2** | 2.8     |
> | 10    | 23.4     | 49.2     | 2.6     |
>
> **Table: Sensitivity of α on WebArena (Avg success rate %)**
>
> | α     | Admin     | GitLab    | Map       | Reddit    | Shopping  |
> | - | - | - | - | - | - |
> | 3     | **53.48** | 39.80     | 35.78     | **58.45** | 40.52     |
> | 5 | 52.31     | **40.78** | **37.66** | 57.64     | **41.67** |
> | 7     | 49.02     | 39.22     | 34.50     | 55.04     | 38.54     |
> | 10    | 46.52     | 37.45     | 33.03     | 53.02     | 36.98     |
>
> Within a reasonable range (α ∈ {3, 5}), performance is robust on both benchmarks. At extreme values (α=10), performance drops — particularly on WebArena, where the structured web tasks benefit less from exploration and excessive randomness hurts.
>
> We sincerely thank the reviewer for the valuable feedback. We will incorporate all the discussions and experimental results above into the revised paper.
>
> [1] Billion-scale similarity search with GPUs
>
> [2] WebRL: Training LLM Web Agents via Self-Evolving Online Curriculum Reinforcement Learning
>
> [3] EvoTest: Evolutionary Test-Time Learning for Self-Improving Agentic Systems

---

> > ### Author Rebuttal · Reviewer_enCW · 2026-04-03
> >
> > Thank you for the rebuttal. My concerns have been addressed.

---

> > > ### Author Response · Authors · 2026-04-06
> > >
> > > Thank you for your kind response — we are truly glad to hear that our rebuttal was able to address your concerns.
> > >
> > > We would also like to assure you that all the points discussed during this review process will be carefully incorporated into the final revision of the paper, further strengthening the work.
> > >
> > > We sincerely hope that our clarifications, and its concerns addressed, may be reflected in your final evaluation. We deeply appreciate the time and thoughtfulness you have devoted to reviewing our work.

---

### Official Review · Reviewer_HYm3 · 2026-03-10

**Soundness:** 3
**Presentation:** 4
**Significance:** 4
**Originality:** 4
**Overall Recommendation:** 5
**Confidence:** 4

**Summary:**

This paper introduces Just-In-Time Reinforcement Learning (JitRL), a training-free framework that enables LLM agents to learn continually without gradient updates. By maintaining a dynamic memory of past experiences, JitRL retrieves relevant trajectories to estimate action advantages on-the-fly, directly adjusting output logits. It achieves state-of-the-art performance on WebArena and Jericho, outperforming fine-tuning methods while reducing monetary costs by over 30 times through efficient test-time policy optimization.

**Compliance With Llm Reviewing Policy:**

Affirmed.

**Final Justification:**

The rebuttal addressed my concerns, and I appreciate the authors' insightful discussion. I will keep my original positive score.

**Key Questions For Authors:**

1. See Weaknesses, how does the proposed method perform in more diverse and complex environments?
2. The proposed method requires a retrieval operation whenever a newer state is encountered. If this approach is extended to general-purpose LLMs, where the generation of each token is treated as a new state, frequent retrieval would be triggered. Will this efficiency concern hinder the extension of JitRL to general-purpose LLMs? *(Kind notice: the response to this question will NOT negatively impact my evaluation.)*

**Limitations:**

yes

**Strengths And Weaknesses:**

**Strengths：**

1. **High Efficiency and Scalability**: This paper addresses a core bottleneck in Reinforcement Learning (RL) by eliminating training overhead. From another perspective, it functions as a form of In-Context Reinforcement Learning (ICRL) but bypasses the common issue of context window explosion. By using non-parametric memory and logit modulation instead of appending experience trajectories into the token context, **it offers a more scalable path for integrating ICRL with LLM agents**.
2. **Simplicity and Practicality**: The framework is elegantly designed and easy to implement. Its "plug-and-play" nature allows it to be **easily applied to frozen, off-the-shelf LLMs**.
3. **Theoretical Proof**: The proposed method is based on a closed-form solution to the KL-constrained policy optimization objective, providing a strong mathematical foundation for its effectiveness.
4. **Good Presentation**: The paper is well-structured and easy to follow.

---

**Weaknesses：**

1. **Limited Benchmark Diversity**: The evaluation is restricted to a relatively small number of benchmarks (WebArena and Jericho). While these are challenging, the paper **lacks validation on more modern and complex agentic tasks**, such as sophisticated tool-use, agentic RAG, or reasoning environments.

---

> ### Author Rebuttal · Authors · 2026-03-30
>
> We sincerely appreciate your time and positive assessment! We address your main concerns below.
>
> > ### Weakness 1 and Question 1: Concern of benchmark diversity.
>
> WebArena is already a complex and realistic agentic benchmark (real websites, DOM interactions), and together with Jericho it covers both structured and free-form environments. Nonetheless, to provide more diverse validation, **we additionally evaluate JitRL on ScienceWorld [1], with tasks spanning physics, chemistry, and biology**. Unlike Jericho (text adventure games) and WebArena (web navigation), ScienceWorld requires scientific reasoning — agents must understand causal relationships (e.g., electrical conductivity, state changes of matter, inclined plane physics) and execute multi-step experimental procedures. The environment features a large action space (around 100 valid actions per step) and provides score-based feedback upon task completion.
>
> **Setup.** We use Gemini-2.5-Flash as the backbone, run 10 episodes per task with 100 steps per episode under the continual learning setting. Due to the rebuttal time constraints, we randomly selected 6 tasks, ensuring coverage across different scientific domains and varying difficulty levels. We compare JitRL against three training-free baselines (EvoTest, AWM, Reflexion, Static, Memory) and will include the full evaluation across all tasks and all baselines in the revised paper.
>
> **Table: ScienceWorld Results (average score)**
>
> | Task | Static | Memory | Reflexion | AWM | EvoTest | JitRL |
> | - | - | - | - | - | - | - |
> | find-plant | 71.40 | 75.00 | 80.00 | 70.00 | 85.00 | **90.80** |
> | test-conductivity | 60.00 | 60.00 | 41.50 | 24.50 | 7.00 | **76.00** |
> | inclined-plane-determine-angle | 40.00 | 40.00 | 14.00 | 17.50 | 39.50 | **71.00** |
> | chemistry-mix | -100.00 | 1.60 | -10.80 | 37.30 | 5.00 | **65.00** |
> | power-component-renewable | 54.70 | 43.00 | 53.20 | 5.50 | 23.50 | **55.00** |
> | freeze | -100.00 | -80.00 | -36.00 | 8.50 | -7.80 | **78.50** |
> | **Average** | 4.35 | 23.27 | 23.65 | 27.22 | 25.37 | **72.72** |
>
> As shown in the table, JitRL achieves the best score compared to baselines.
>
> > ### Question 2: Will frequent retrieval hinder extension to general-purpose LLMs where each token is a new state?
>
> Thank you for this insightful question. Extending JitRL to general-purpose LLMs (where each token is a decision point) poses several challenges: the token-level state space is nearly infinite, making it difficult to define meaningful "similar states" via text; the action space is the entire vocabulary (~100K), making per-token Q-value estimation infeasible; and the contribution of individual tokens is hard to quantify through rewards.
>
> A promising direction to address this is **embedding-based state representation**: instead of using text to represent states, we can use the LLM's hidden state embeddings as dense state vectors. These embeddings naturally capture the semantic context at each generation step, and similarity retrieval in embedding space is well-supported by efficient approximate nearest-neighbor search (e.g., Faiss [2], which can **search billions of vectors in milliseconds**). This would preserve JitRL's core framework — non-parametric retrieval → advantage estimation → logit adjustment — while extending it beyond text-based state matching. We leave the exploration of embedding-based state representation as future work.
>
> We sincerely thank the reviewer again for the constructive feedback. We will incorporate these discussions along with the complete ScienceWorld experiments into the revised paper.
>
> [1] ScienceWorld: Is your Agent Smarter than a 5th Grader?
>
> [2] Billion-scale similarity search with GPUs

---

> > ### Author Rebuttal · Reviewer_HYm3 · 2026-04-03
> >
> > Thanks for the authors' rebuttal. The additional experiments addressed my concerns. I appreciate the authors' insightful discussion of a potential path to extend JitRL to general-purpose LLMs. I will keep my original positive score. Thank you and good luck!

---

> > > ### Author Response · Authors · 2026-04-06
> > >
> > > Thank you again for your thoughtful and encouraging feedback! We are truly glad that the additional experiments were helpful and that our discussion resonated with you.
> > >
> > > As the discussion phase comes to a close, we would be sincerely grateful for any continued support you may be able to offer. Your positive engagement with our work has meant a great deal to us.
> > >
> > > Thank you again for your time and generosity throughout this process. We wish you all the best as well!

---

### Official Review · Reviewer_AtJL · 2026-03-12

**Soundness:** 3
**Presentation:** 2
**Significance:** 2
**Originality:** 4
**Overall Recommendation:** 5
**Confidence:** 4

**Summary:**

This paper proposes Just-In-Time Reinforcement Learning (JitRL), a training-free test-time adaptation method for LLM agents that stores past (state, action, return) tuples, retrieves similar experiences online, estimates local state/action values, and shifts candidate-action logits via a KL-regularized closed-form update. The method is instantiated with LLM-based stepwise reward assignment and task-specific state abstractions, and evaluated on WebArena and Jericho with additional transfer, ablation, and cost studies. The reported results show consistent gains over training-free baselines and favorable cost/performance relative to selected weight-update baselines on weaker models.

**Compliance With Llm Reviewing Policy:**

Affirmed.

**Final Justification:**

The extensive additional experiments performed during the rebuttal period addressed my concerns and questions. I have therefore increased my score and recommend acceptance.

**Key Questions For Authors:**

* Can you provide same-backbone, larger budget, multi-seed comparisons against WebRL/GRPO on the main benchmarks? If the gains remain, my confidence in the central empirical claim would increase substantially.
* How sensitive are results to the LLM evaluator that assigns stepwise rewards, and to the specific state abstractions/retrieval heuristics?
* Did you ablate the decision of assigning an optimistic value to less frequently seen actions?
* Would you expect $k$ having to increase as tasks become more complex since advantage estimation might require more samples?
* Can you disclose your hyperparameter sweep for GRPO? Which learning rates / batch sizes did you try?


Typos:
* I believe $\hat{A}$ in Eq. (10) should instead be $A$ to be consistent with the surrounding equations.

**Limitations:**

Partly yes. However, in my view the paper slightly overreaches in its current claims that JitRL is advantageous over training-based RL. I believe remaining limitations of the empirical evaluation should be discussed (as mentioned above).

**Strengths And Weaknesses:**

Strengths:
* The paper addresses an important problem: continual adaptation of frozen LLM agents. The core idea is clean and practically appealing: retrieval-based advantage estimation plus direct logit modulation, which is more principled than pure prompt-memory approaches.
* The paper is generally clear, with helpful figures/tables, and the empirical section is broad: two benchmarks, backbone transfer, cross-task transfer, qualitative analyses, extensive ablations, and reporting evaluation costs.
* The paper includes a simplified theoretical analysis that supports the principled framework.

Weaknesses:
* The headline claim of outperforming weight-update RL is not fully convincing because the main comparisons from the main body are not controlled: WebArena uses different backbones than WebRL/SFT, Jericho compares to task-specific GRPO checkpoints on another base model, and the same-backbone comparison in Appendix I is smaller and more mixed. Additionally, the same-backbone comparison only evaluates a low-sample setting. Arguably, training-based methods tend to require more samples to match performance of training-free methods. An analysis of this distinction would be interesting.
* Additionally, in my view, the paper lacks an adequate discussion of the limitations of training-free/non-parametric learning. It remains to be seen to what extent fully training-free methods can scale to more complex domains with more data, collected over longer time horizons.
* The practical gains may depend heavily on task-specific engineering (e.g., regularized URLs and retrieval/state abstractions for WebArena, structured summaries for Jericho, and an LLM evaluator for stepwise rewards) so the claim of a broadly general continual-learning framework feels somewhat overstated.
* The paper does not compare against more recent reflection-based baselines such as ACE [1]

---

[1]: Zhang et al. Agentic Context Engineering: Evolving Contexts for Self-Improving Language Models. 2025.

---

> ### Author Rebuttal · Authors · 2026-03-30
>
> Thank you for recognizing JitRL's novelty! We address your main concerns as follows:
>
> > ### W1 & Q1: Concern of the comparison between JitRL and RL methods.
>
> For WebArena, we directly evaluate the released SFT and WebRL checkpoints. For Jericho, we use GRPO baseline following EvoTest [2]. The same-backbone comparison in Appendix I **uses a low-sample setting because we target continual learning regime**, as stated in our title and introduction. Here we also provide a same-backbone controlled comparison on Jericho:
>
> 📊 **[Table: Comparison on Jericho with same backbone](https://imgur.com/a/UtYQy3i)**
>
> To reach the full potential of training-based methods, we conduct **a new controlled, larger-budget experiment** same as WebRL, using the same backbone (Llama-3.1-70B-Instruct), the same set of training tasks (WebArena tasks excluding WebArena-Lite), and the **same protocol with separate training and evaluation phases**. During training, JitRL collects experience memory (no gradient updates); during evaluation, JitRL is tested on WebArena-Lite using only the pre-collected memory.
>
> 📊 **[Table: Comparison on same backbone, with larger budget](https://imgur.com/a/WZCHJ9t)**
>
> As shown in the table, JitRL significantly outperforms SFT and is only 4.97% behind WebRL, at ~\$200 cost vs. WebRL's ~\\$9,900 cost. Note that **JitRL is specifically designed for continual learning** where abundant offline training data is unavailable. We will add these controlled comparisons to the main paper.
>
> > ### W2: Limitations of training-free methods and their scalability.
>
> We believe training-free and training-based methods are complementary paradigms suited to different regimes:
>
> Training-free methods are preferred when: (1) only limited samples are available — training-free methods tend to be more sample-efficient than gradient-based RL in this regime, a finding also echoed by GEPA [1]; (2) training compute is unavailable or expensive; (3) one wants to leverage powerful frozen API models.
>
> Training-based methods are preferred when: (1) the task requires knowledge that is hard to state in language — e.g., time-series forecasting patterns or spatial reasoning (chess board evaluation), where parametric training can encode non-verbal patterns into weights but textual memory retrieval cannot; (2) sufficient training data is available and the task distribution is fixed (i.e., no need for continual adaptation).
>
> Regarding scalability: as shown in the table below, **JitRL's performance continuously improves as memory grows, while retrieval overhead remains negligible compared to LLM inference time**. More details can be found in our response to `Reviewer enCW (W2 & W3)`.
>
> 📊 **[Table: Scalability](https://imgur.com/a/NtRO3q9)**
>
> > ### W3 & Q1: Concern about task-specific engineering.
>
> We built a **unified pipeline** (same state representation, retrieval, and evaluator for both benchmarks). As shown in the table below, **JitRL still achieves nearly the same performance and outperforms all baselines**, confirming that gains primarily stem from the advantage estimation and logit update mechanism. More details can be found in our response to `Reviewer enCW (W1)`.
>
>
> 📊 **[Table: Unified Pipeline](https://imgur.com/a/uLDu83N)**
>
> **LLM evaluator sensitivity.** Table 4 in our paper shows consistent gains across three different LLM backbones (also serving as evaluators), suggesting **robustness to evaluator choice**.
>
> > ### W4: Comparison with ACE.
>
> We have compared ACE with JitRL on all three Jericho games.
>
> 📊 **[Table: ACE vs JitRL](https://imgur.com/a/elAqTry)**
>
> As shown in the table, JitRL outperforms ACE. We will include ACE as a baseline in the revision.
>
> > ### Q3: Ablation of the optimistic value for unseen actions
>
> We ablated both parameters of the optimistic exploration bonus (Eq. 6) on both benchmarks:
>
> 📊 **[Table: λ ablation](https://imgur.com/a/MT6Zkou)**
>
> 📊 **[Table: α ablation](https://imgur.com/a/tkwtUur)**
>
> As shown in the tables, performance is stable across a reasonable range of both parameters. WebArena favors a smaller λ since less exploration is needed, while Jericho benefits from a larger λ.
>
> > ### Q4: Would k need to increase as tasks become more complex
>
> Yes, more complex tasks typically involve more states and actions, so a larger k may be needed to provide sufficient coverage for reliable advantage estimation.
>
> > ### Q5: Hyperparameter sweep for GRPO
>
> Our GRPO directly follows EvoTest [2]'s configuration. Learning rate = 1e-6, batch size = 1 (updating after each episode, aligned with JitRL's continual learning setting), with 8 rollout per prompt.
>
> > ### Typo: Eq. (10)
>
> The use of Â in Eq. (10) is intentional: Eqs. (8)–(9) use the theoretical optimal advantage A, while Eq. (10) uses the estimated Â from memory retrieval. We will clarify this in the revision.
>
> [1] GEPA: Reflective Prompt Evolution Can Outperform Reinforcement Learning
>
> [2] EvoTest: Evolutionary Test-Time Learning for Self-Improving Agentic Systems

---

> > ### Author Rebuttal · Reviewer_AtJL · 2026-04-03
> >
> > Thanks for your detailed rebuttal and running additional experiments. All my concerns beyond the hyperparameter sweep for GRPO are addressed. I would encourage the authors to include the results with larger budget in the main paper.
> >
> > Regarding the hyperparameters of GRPO, I'm not very familiar with EvoTest, however, after a brief skim of the paper I didn't find how they selected their hyperparameters. Can the authors point me to the place where they describe their hyperparameter sweep?
> >
> > Relatedly, I think it might be good to compare to GRPO with a larger batch size than 1 (e.g., 16 or 32). While larger batch sizes may learn more slowly over very small samples, updates may be more stable with larger batch sizes. An additional axis with larger batch sizes is the mini batch size (often ~1/4-th of the batch size) and ppo epochs. Pointing to an existing hyperparameter sweep in EvoTest or performing one would significantly strengthen my confidence in the paper's results.
> >
> > As mentioned, I recognize the paper's novelty and contribution, and would be happy to increase my score if this final concern is addressed.

---

> > > ### Author Response · Authors · 2026-04-06
> > >
> > > Thank you for your constructive feedback. We have conducted the hyperparameter sweep and present the results below.
> > >
> > > ### **Hyperparameters Swept**
> > >
> > > Following your suggestion, we swept four hyperparameters: **rollout count** $n \in \\{8, 16, 32\\}$, **batch size** $\in \\{1, 8\\}$, **PPO epochs** $\in\\{ 1, 2, 4\\}$, and **learning rate** $\in \\{1\text{e-}6, 1\text{e-}5\\}$. For each experiment, we vary one axis at a time while fixing the others.
> > >
> > > ### **Clarification on Batch Size vs. Rollout Count**
> > >
> > > We would like to clarify the distinction between rollout count and batch size, which is particularly relevant in our setting where the training data consists of **a single problem instance** (analogous to a single prompt in standard GRPO — here, the initial game observation from which all trajectories begin).
> > >
> > > In GRPO, each problem instance is rolled out $n$ times. The advantage for each trajectory is computed via group-relative normalization within trajectories sharing the same instance:
> > >
> > > $$A_i = \frac{r_i - \text{mean}(r_1, \dots, r_n)}{\text{std}(r_1, \dots, r_n)}$$
> > >
> > > Since all our data originates from the same problem instance, increasing rollout count and increasing batch size both increase the total number of trajectories per step. In both cases, the final policy gradient is averaged over all trajectories. The difference lies in the **scope of group normalization**:
> > >
> > > - **Increasing rollout count $n$** (e.g., $n{=}64$, batch size$=$1): All 64 trajectories share the same group ID, so group-relative normalization is performed over all 64 trajectories jointly. This yields a smoother advantage estimate.
> > > - **Increasing batch size** (e.g., $n{=}8$, batch size$=$8): Each copy of the instance in the batch receives an independent group ID, creating 8 independent groups of 8 rollouts. Group-relative normalization is performed separately within each group of 8. This makes the advantage more sensitive to local outliers within each group.
> > >
> > > ### **Experimental Setup**
> > >
> > > We conduct experiments on Zork1 using Qwen3-32B (bf16) trained with GRPO for 50 training steps, matching the number of episodes used in JitRL evaluation. Note that JitRL uses exactly **one trajectory** per step, whereas GRPO requires **rollout × batch_size** trajectories per step. After each training step, we evaluate the model with a single rollout and record the score. We use these scores to compute the mean validation score and plot the 50-step learning curve. All experiments are run on 4× NVIDIA H200 GPUs with tensor_parallel=4, with a max of 80 turns per episode, max response length of 8,192 tokens, and KL coefficient of 0.001 (low_var_kl).
> > >
> > > ### **Results**
> > >
> > >
> > > | rollout | batch size | ppo_epochs | lr       | Val Mean | Val Max | Val Min |
> > > | ------- | ---------- | ---------- | -------- | -------- | ------- | ------- |
> > > | 8  | 1          | 1          | 1e-6     | 16.2     | 35      | 0       |
> > > | 16  | 1          | 1          | 1e-6     | 13.6     | 40      | 0       |
> > > | 32 | 1          | 1          | 1e-6     | 12.0     | 40      | -5      |
> > > | 8       | 1          | 2     | 1e-6     | 10.4     | 40      | 0       |
> > > | 8       | 1          | 4     | 1e-6     | 17.1     | 40      | 5       |
> > > | 8       | 1          | 4          | 1e-5 | 25.5     | 45      | -5      |
> > > | 8       | 8     | 1          | 1e-5 | **40.7** | **55**  | 5       |
> > >
> > >
> > > We also provide the 50-step learning curves for all GRPO configurations alongside JitRL (all using Qwen3-32B) in:
> > >
> > > 📊 [Figures](https://imgur.com/a/4LW8PKt)
> > >
> > > ### **Analysis**
> > >
> > > The best configuration (rollout=8, batch_size=8, ppo_epochs=1, lr=1e-5) achieves a mean validation score of **40.7** and a maximum of **55**. Notably, the validation score shows a clear upward learning curve, stabilizing at 55 in the final 20 steps. This is similar to JitRL's mean score of 42.1 on Zork1 using the same Qwen3-32B model.
> > >
> > > However, **this result comes at a substantial sample cost**. Over 50 steps, JitRL requires exactly **50 trajectories** in total. In contrast, GRPO requires **50 × rollout × batch_size** trajectories — each training step in this best configuration alone consumes 64 trajectories (8 rollouts × 8 batch size), **exceeding JitRL's entire 50-episode budget**. In total, this configuration consumes **3,200 training episodes** — over 60× more than JitRL. Even our original paper's GRPO setting (rollout=8, batch_size=1) requires **400 training episodes**, 8× more than JitRL. While GRPO with extensive tuning can approach or exceed JitRL's performance, this comes at a substantially higher sample cost.
> > >
> > > We thank you for your time and constructive feedback. We will update the paper to include the GRPO training code, detailed configuration, the full hyperparameter sweep, and the corresponding results.

---

### Decision · Program_Chairs · 2026-04-30

**Decision:**

Accept (spotlight)

**Comment:**

The paper addresses an important problem: continual learning with RL
in LLMs.  The idea of storing experience and using it to estimate
advantage and reweight logits is appealing and well explained.  The
simple theoretical analysis connecting it to KL-regularized RL
strengthens the contribution of the paper and improves understanding.
The weakest point of the paper was limitations in the empirical study,
particularly missing same-backbone comparisons.  These were resolved
during the rebuttal and additional experiments were added leading to
increased scores and an accept recommendation.